# Identification of key genes involved in secondary metabolite biosynthesis in *Digitalis purpurea*

**Fatemeh Amiri, Ali Moghadam** (ORCID)**\*, Ahmad Tahmasebi, Ali Niazi**

Institute of Biotechnology, Shiraz University, Shiraz, Iran

\* ali.moghadam@shirazu.ac.ir

**Data Availability Statement:** All relevant data are within the paper and its Supporting information files.

**Funding:** The author(s) received no specific funding for this work.

## Abstract

The medicinal plant *Digitalis purpurea* produces cardiac glycosides that are useful in the pharmaceutical industry. These bioactive compounds are in high demand due to ethnobotany's application to therapeutic procedures. Recent studies have investigated the role of integrative analysis of multi-omics data in understanding cellular metabolic status through systems metabolic engineering approach, as well as its application to genetically engineering metabolic pathways. In spite of numerous omics experiments, most molecular mechanisms involved in metabolic pathways biosynthesis in *D. purpurea* remain unclear. Using R Package Weighted Gene Co-expression Network Analysis, co-expression analysis was performed on the transcriptome and metabolome data. As a result of our study, we identified transcription factors, transcriptional regulators, protein kinases, transporters, non-coding RNAs, and hub genes that are involved in the production of secondary metabolites. Since jasmonates are involved in the biosynthesis of cardiac glycosides, the candidate genes for *Scarecrow-Like Protein 14* (*SCL14*), *Delta24-sterol reductase* (*DWF1*), *HYDRA1* (*HYD1*), and Jasmonate-ZIM domain3 (*JAZ3*) were validated under methyl jasmonate treatment (MeJA, 100 μM). Despite early induction of *JAZ3*, which affected downstream genes, it was dramatically suppressed after 48 hours. *SCL14*, which targets *DWF1*, and *HYD1*, which induces cholesterol and cardiac glycoside biosynthesis, were both promoted. The correlation between key genes and main metabolites and validation of expression patterns provide a unique insight into the biosynthesis mechanisms of cardiac glycosides in *D. purpurea*.

## Introduction

The human has implemented medical plants for therapeutic purposes or disease prevention with the purpose of surviving since ancient times [1–4]. Most of the industries, specifically the pharmaceutical industry, implement secondary metabolites [5]. According to the estimations, 25% of prescribed drugs in industrialized nations contain natural plant products [6]. Inkwood research website reported that the plant-derivative drug turnover was $26,621 million in 2017; also, it is estimated to be $53,850 million by 2026 [7]. Due to the lack of accurate information regarding the molecular mechanisms and gene networks involved in the

**Competing interests:** The authors have declared that no competing interests exist.

regulation of metabolic pathways in most of the medicinal plants, researchers discover the proposed biological pathways-involved target genes through integrative omics data such as comparative co-expression analysis [8, 9]. Despite the clinical and chemical significance of the above-mentioned plants, there is not a comprehensive study on metabolite biosynthesis; therefore, characterization of key genes involved in secondary metabolites synthesis could be informative and practical to improve the yield of valuable metabolites.

Specifically, secondary metabolites are produced in certain plant species. The extraction and refining of these products within specialized cells is difficult at certain stages of the plant life cycle under specific conditions and in small amounts [10, 11]. Key genes that affect the whole network and increase the secondary metabolites amounts and production pathways [9] could be detected using systems biology approaches [12, 13]. Therefore, the integrative omics approach is an efficient biological analysis of systems [12, 13] that leads to a better understanding of transcriptome and metabolome interactions, as well as valuable secondary metabolites-related gene networks [9, 14, 15]. The gene networks key genes could be detected through various functional genomics studies, such as Quantitative Trait Locus (QTL), Genome-Wide Association Studies (GWAS), forward and reverse mutagenesis screens, targeted mutagenesis approaches, and omics techniques [11, 13]. Using the co-expression network analysis, Tahmasebi et al. (2018) identified important modules associated with the secondary metabolism in *Echinacea purpurea* [14]. The co-expression analysis is widely applied to the biosynthesis of secondary metabolites [16–19], stem lodging resistance regulation in *Brassica napus* [20], identification of critical modules and candidate genes of drought resistance in *Triticum aestivum* [21], regulation of flower and fruit development in *Fragaria vesca* [22], and identification of genes involved in resistance responses to powdery mildew in *Hordeum vulgare* L. var. *nudum* [23].

Biological networking is considered as an appropriate approach to detect the metabolic pathways and resources involved in the production of secondary metabolites [10, 24]. According to the investigations on various organisms, genes with similar functional roles tend to be co-expressed [24]. In most of the cases, including non-model organisms, the co-expression analysis is a straightforward approach to predict the gene functions [24, 25]. Weighted Gene Co-expression Network Analysis (WGCNA) is frequently applied to detect modules with similar expression profiles, which can be included in similar biological processes or pathways [24, 26, 27]. In other words, the functionally-related genes tend to be co-expressed [22, 24] and the mentioned method is a powerful tool to identify the correlation between gene expression profiles and phenotypes, as well as new metabolic pathways in plants [27]. Current study applied the co-expression analysis to identify hub genes with secondary metabolites biosynthesis, including the biosynthesis of dioscin in *Dioscorea nipponica* [28], monoterpenoids, fatty acid derivatives, isoflavonoids, and anthocyanins biosynthesis in *Echinacea purpurea* [14], tanshinone biosynthesis in *Salvia miltiorrhiza* [16], terpenoid biosynthesis pathways in *Matricaria recutita* and *Chamaemelum nobile* [17], as well as flavonoid metabolism regulation in *Camellia sinensis* [19].

*Digitalis purpurea* (L.) (Foxglove, Common Foxglove, and Purple Foxglove) is a medicinal and ornamental plant [6, 29] that produces valuable bioactive metabolites, such as cardiac glycosides (acetyldigitoxin, digitoxin, digoxin, gitaloxigenin, gitaloxin, gitoxin, purpurea glycoside A, purpurea glycoside B, and strospeside), flavonoids (cyaniding and digicitrin), anthraquinones, saponins (digitonin) and phenylethanoid glycosides (calceolarioside, cornoside, forsythiaside, plantainoside, plantamajoside, and sceptroside) [6, 29]. Purpurea glycoside A (glucodigitoxin) and purpurea glycoside B are the principal glycosides in the fresh leaves, which are converted into digitoxin and gitoxin respectively, which normally predominate in the dried leaf [30]. Gitoxin plays an important role in treating breast cancer [6].

Evatromonoside, is precursor of digitoxigenin bis-digitoxoside while digitoxigenin bis-digitoxoside is precursor of digitoxin which has been used as cardiac drug [31].

Digitoxin as an in-use medication has anti-HSV (herpes simplex virus type) activity which actively inhibits HSV-1 replication [6]. In addition, it suppresses hypersecretion of IL-8 from cultured CF (cystic fibrosis) lung epithelial cells [6]. It is worth noting, digitoxin could induce apoptosis in tumor cells [6, 29]. Therefore, digitoxin could be a novel drug class antiviral mechanism and a candidate drug for suppressing IL-8-dependent lung inflammation in CF and potential anticancer drugs [6, 29]. The pharmacological activity of the glycosidal extract of *D. heywoodii* is related to gitoxin derivatives (digitalinum verum and strospeside) [32].

Generally, cardiac glycosides are efficiently applied to treatments for heart diseases and cancers [6, 29]; moreover, most of these compounds biosynthesis-involved gene functions are unknown. To achieve a better understanding of specific molecular mechanisms of secondary metabolism-related genes of this medicinal plant, the correlation between transcriptome and metabolome was studied using WGCNA. In addition, the major modules and hub genes were identified. Eventually, other novel agents correlated with the biosynthesis of secondary metabolites, such as transcription factors (TFs), transcriptional regulators (TRs), protein kinases (PKs), transporters, and mRNA-like non-coding RNAs (mlncRNAs) were identified through the functional analysis. To confirm the integrative data analysis, the candidate genes were validated after methyl jasmonate (MeJA) treatment.

## Materials and methods

### Data collection and preprocessing

Fig 1 represents the data collection and preprocessing steps (Fig 1). The normalized RNA-Seq data of eight different tissues of *D. purpurea*, including mature flower, immature flower, sepals mature flower, sepals immature flower, immature leaf, young leaf, mature leaf petiole, and young leaf petiole, were retrieved from the Medicinal Plant Genomics Resource database (http://medicinalplantgenomics.msu.edu/) [33]. In addition, metabolomic information of major metabolites, such as digitoxigenin bis-digitoxoside ($C_{35}H_{54}O_{10}$), digitoxin ($C_{41}H_{64}O_{13}$), gitoxin ($C_{41}H_{64}O_{14}$), glucodigitoxin ($C_{47}H_{74}O_{18}$), and strospeside ($C_{30}H_{46}O_9$), within the tissues, which were in accordance with transcriptome, was retrieved from the Plant/Eukaryotic and Microbial Systems Resource [34] (S1 Table). According to this database, liquid chromatography/time-of-flight/mass spectrometry (LC/TOF/MS) method was applied to achieve the metabolome profiles. First, the transcript expression level and the quantity of secondary metabolites within eight plant tissues of transcriptomic and metabolomic data were classified correspondently according to a comparative analysis using R Package WGCNA. Transcripts with low expression values observed in the transcriptomic data were also filtered out using the genefilter package based on the variance filtering through varFilter function.

### Gene co-expression network construction

Gene co-expression networks were achieved through the WGCNA package in R space [35]. After transcripts filtration with low expression; the expression filtered matrix was included in the WGCNA workflow. The scale-free topology criterion was implemented to determine the soft threshold power, which is defined as the similarity relationships between gene pairs by calculating the unsigned Pearson's correlation matrix [36]. The network was constructed using a step-by-step network construction method, which was on the basis of adjacency matrix construction and its consequent turning into the topological overlap matrix (TOM) that it was applied with the purpose of describing the interconnection among genes and finally calling the hierarchical clustering function [36]. Next, modules with a minimum module size of 30 were

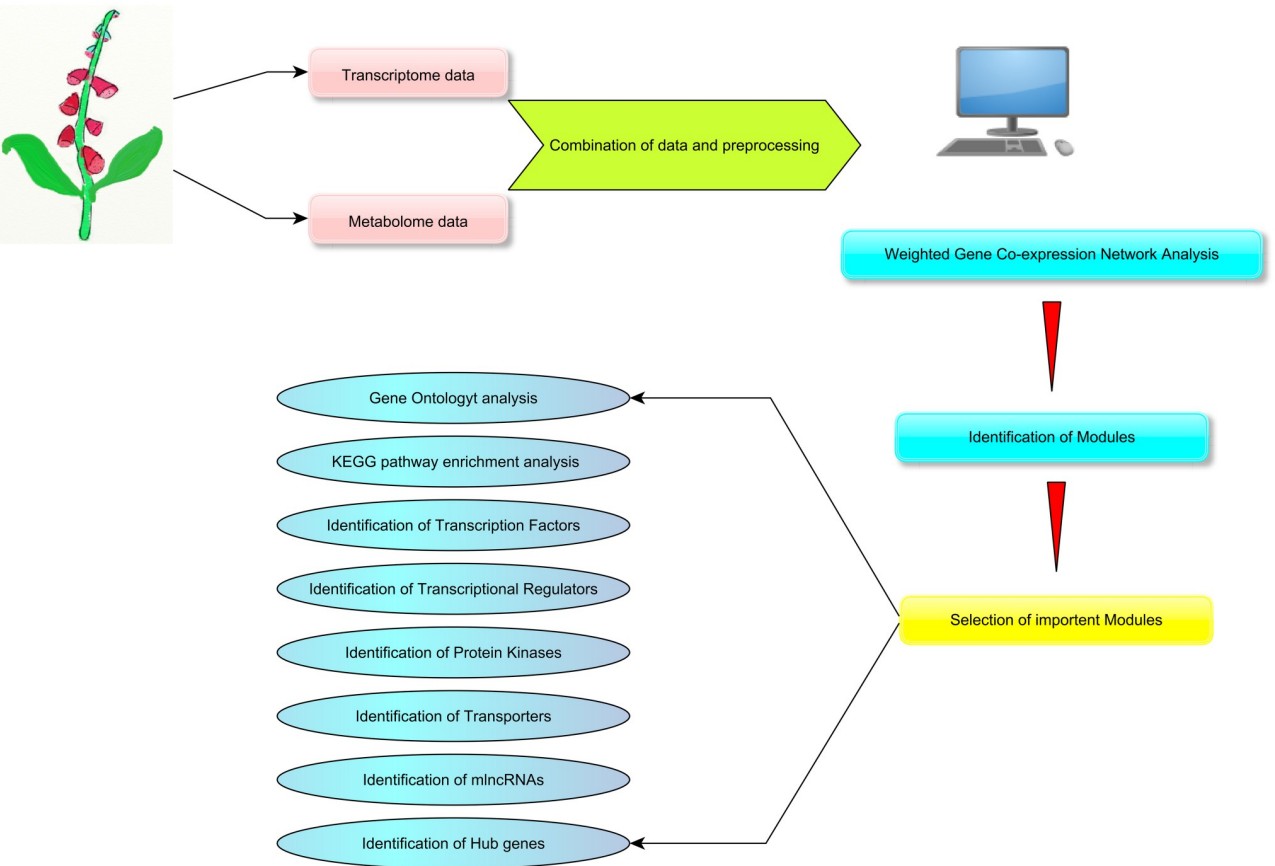

**Fig 1. Flowchart of gene co-expression network analysis in *D. purpurea*.** Data collection and analysis towards downstream analyses are shown.

identified, and then, those with similar expression profiles were merged through the Dynamic Tree Cut method with a CutHeight of 0.25 [35]. Finally, the eigengene values of the modules were calculated to evaluate the module-trait (secondary metabolites) relationships and detect the most significant association of the modules to the secondary metabolites [36]. To quantify the associations of individual genes with traits of interest (digitoxigenin bis-digitoxoside), the correlation between individual genes and the trait (digitoxigenin bis-digitoxoside) was defined as Gene Significance (GS) [37]. In addition, a quantitative measure of module membership (MM) and gene expression profile were defined with the purpose of quantifying how close a gene is to a given module [35]. Generally, if GS and MM were highly correlated, it would imply that genes were the highly important elements for the modules and were most significantly associated with the trait [37, 38]. Finally, genes with a high significance for digitoxigenin bis-digitoxoside and high MM in chocolate3 module were identified.

## Eigengene network visualization through WGCNA functions

A plot summary of the eigengene network was generated according to the plotEigengeneNetworks convenient function [35]. The trait (digitoxigenin bis-digitoxoside) was added to eigengenes with the purpose of understanding how the trait fit into the eigengene network. In fact, a sample trait, such as digitoxigenin bis-digitoxoside, could be incorporated as an additional node of the eigengene network [35]. The adjacency between the sample trait and an eigengene

sometimes could be considered as the eigengene significance [35]. Therefore, we evaluated the relationship between each module and digitoxigenin bis-digitoxoside by correlating the eigengenes for each module to digitoxigenin bis-digitoxoside and consequently, the eigengene dendrogram and heatmap identified groups of correlated eigengenes called meta-modules [35]. Module-module relationship, which is also called meta-module, is a group of correlated eigengenes with the correlation of eigengenes of at least 0.5 [35, 37]. Meta-modules are defined as tight clusters of modules [35, 37] and groups of highly correlated eigengenes [39].

### Functional annotation and enrichment analysis

To convert the transcript sequences to the orthologues, BLASTX with $E\text{-}value \leq 10^{-5}$ was applied against the Arabidopsis Information Resource (TAIR). The functional enrichment analysis of modules was performed using Database for Annotation, Visualization and Integrated Discovery (DAVID) [40] for categories of Biological Process (BP), Molecular Function (MF), and Cellular Component (CC). Kyoto Encyclopedia of Genes and Genomes (KEGG) pathway enrichment analysis was also carried out in the web-based DAVID [40]. $P\text{-}value < 0.01$ was considered to be significant; moreover, the identification and classification of TFs, TRs, and PKs were carried out through applying the transcript sequences to BLASTX search against the iTAK database [41]. To identify transporters, BLASTX was carried out on transcript sequences against the transporter classification database (TCDB) with $E\text{-}value \leq 10^{-20}$ [42]. Wu et al. (2012) identified 2660 mlncRNAs candidates, which were considered as an emerging class of regulators, using a computational mlncRNA identification pipeline in *D. purpurea* [43]. After the creation of the mlncRNAs-derived local database through CLC Genomics Workbench 11.0, the searching procedure was carried out for all of the transcripts in the significant major modules using BLASTN with a cut-off $E\text{-}value \leq 10^{-5}$ to uncover important mlncRNAs.

### Identification of hub genes

The most efficient genes of the selected modules were detected through the computational analysis of the connectivity between nodes. Using the CytoHubba plugin of Cytoscape software with Maximal Clique Centrality (MCC) function, the edge count-based identification of the top 20 hub genes was carried out in each selected module.

### Plant materials and growth conditions

Two-year-old *D. purpurea* were purchased from the Sepahan flower & ornamental plants international market (Isfahan, Iran). The plants that were in the flowering phase were placed under a 16-h light: 8-h dark cycle at 25˚C for two weeks to be compatible with the new conditions in the glasshouse (S1 Fig).

### Stress conditions

The adopted plants were sprayed and watered with the solution of 100 μM MeJA (plus 0.1% Tween-20) in 0.1% ethanol. The control plants also were sprayed and watered with 0.1% Tween-20 in 0.1% ethanol. All the pots were covered with the plastics. The leaf samples were collected at 3, 6, 24, and 48 hours after treatment (S1 Fig). The collected leaves were immediately frozen in liquid nitrogen and stored at −80˚C until used for RNA extraction.

## RNA extraction, DNase treatment, and cDNA synthesis

Total RNA was extracted from leaf samples using the Column RNA Isolation Kit (DENAzist Asia, Mashhad Iran) according to the manufacturer's instructions. The quantity and concentration of RNA was measured using a NanoDrop ND 1000 Spectrophotometer (Thermo Fisher Scientific, Wilmington, DE, USA). The integrity and quality of RNA was checked by visual observation of 28S and 18S rRNA bands on a 1% agarose gel. Prior to use, RNA samples were stored at −80˚C. DNase treatment of RNA was carried out using the RNase-free DNase kit (Thermo Fisher Scientific, Waltham, Massachusetts, USA) according to the manufacturer's instructions. The quality and quantity of treated RNA were rechecked by NanoDrop and agarose gel respectively. Then, 1 µg of DNase-treated RNA was used for first-strand cDNA synthesis using SinaClon BioScience kit (Karaj, Iran) according to the manufacturer's instructions. The cDNA samples were stored at −20˚C prior to use.

## Candidate genes

Based on integrative data analysis, four following genes were candidates for the validation. *Jasmonate-ZIM domain3* (*JAZ3*) was identified in the coral3 module that showed a high positive correlation and significant *P-value* with glucodigitoxin and strospeside (S2 Table). *JAZ3* was chosen to know whether it responds to MeJA induction or not. The JAZ family genes are key repressors in the JA signal transduction pathway. *Scarecrow-Like Protein 14* (*SCL14*) is a hub gene in the blue2 module and interacts with TGA II TFs including TGACG sequence-specific binding protein 2 (TGA2), Ocs-element-binding factor 5 (TGA5), and TGACG motif-binding factor 6 (TGA6) that affects the transcription of stress-responsive genes. It was associated with the secondary metabolites of digitoxigenin bis-digitoxoside and gitoxin (S2 Table). *Delta24-sterol reductase* (*DWF1*) is *SCL14* downstream gene that is involved in the steroid biosynthesis (S2 Fig). This enzyme was identified in the coral3 module that showed a high positive correlation and significant *P-value* with glucodigitoxin and strospeside (S2 Table). *HYDRA1* (*HYD1*) was identified in the darkorange2 module, which significantly was associated with digitoxigenin bis-digitoxoside and was involved in the process of steroid biosynthesis (S2 Fig and S2 Table). In fact, *HYD1* converts 4α-methylcholesta-8,24-dien-3β-ol to 4α-methylcholest-7,24-dien-3β-ol and *DWF1* converts desmosterol to cholesterol (precursor of cardiac glycosides) in the steroid biosynthesis pathway. The steroid biosynthesis pathway is one of the putative biosynthetic pathways of plant cardiac glycosides.

## Primer design

Primers were designed using Allele ID 7 and Vector NTI 11 software for the reference and candidate genes (Table 1). The primers were designed based on the aligned nucleotide file. In this project to be more precise, we quantified the final expressions based on the means of reference gene of actin and other four genes.

**Table 1. Primer sequences used in the study.**

| Gene | Forward | Reverse | Accession number | Ta (˚C) | PCR product (bp) |
|---|---|---|---|---|---|
| *DWF1* | CTTCTCACTCTTGCGACCTT | GGATTCCAGCCACACACT | *AT3G19820* | 55 | 156 |
| *SCL14* | CGCTGTTCCACTTCTCCGCCAT | CCTGCCACTGCTTGTATGTCTCT | *AT1G07530* | 61 | 172 |
| *HYD1* | GAACCCTCATTTCCTTGCCGAAGT | CCCAAAGACACGCCAAACTGAAGA | *AT1G20050* | 55 | 195 |
| *JAZ3* | GTCGGTGTGCGTGTATGA | ATGGATGCTGGAACTGGC | *AT4G34990* | 62 | 123 |
| *Actin* | GTCTCTCACAATTTCCTTCTCAG | GCTCTCCCACACGCTATT | *AT2G37620* | 55 | 126 |

### RT-qPCR to validation of candidate genes

First, primer specificity was confirmed by PCR and sequence analysis. To minimize pipetting error, the cDNA samples were diluted 1:20 by using nuclease-free water, and 5 μL cDNA was used for RT-qPCR. Relative RT-qPCR was performed in a 20 μL volume containing 5 μL cDNA (diluted), 10 μL RealQ Plus 2x Master Mix Green (Sinuhebiotech, Shiraz, Iran), 0.7 μL of 10 μM primers (10 μM). The amplification reactions were carried out in a Line-gene K thermal cycler (Bioer, China) under the following conditions: 15 s at 95˚C, 45 cycles of 94˚C for 15 s, Ta temperature for 15 s, and 72˚C for 20 s. After 45 cycles, the specificity of the amplifications was tested by melting curve analysis by heating from 50 to 95˚C. All amplification reactions were repeated two times under identical conditions and included a negative control and 3 standard samples.

### RT-qPCR data analysis

The relative expression of candidate genes was calculated based on the threshold cycle ($C_T$) method. The $C_T$ for each sample was calculated using the Line-gene K software. When replicate PCRs are run on the same sample, it is more appropriate to average $C_T$ data before performing the $2^{-\Delta\Delta CT}$ calculation. The actin gene was used as the reference gene for data normalization. The determined mean $C_T$ values for the candidate and internal reference gene were used in equation $2^{-\Delta\Delta CT}$ = ($C_{T, \text{candidate genes}}$ — $C_{T, \text{housekeeping genes}}$) Time x — ($C_{T, \text{candidate genes}}$ — $C_{T, \text{housekeeping genes}}$) Time x [44]. Time x represents the expression of the candidate genes at any time point in control and treated plants. The fold change ratios of the genes were normalized to internal control genes and were calculated relative to the expression at any time in control plants.

### Statistical analysis

Analysis of variance followed by Duncan's multiple range test was performed using MINITAB (Minitab, Inc., Pennsylvania, USA). In all cases, differences were regarded to be statistically significant at *P-value* $\leq$ 0.05 level. All experiments were performed in triplicate, analyzed using the GraphPad Prism software (GraphPad Software, USA).

## Results and discussion

### Co-expression network analysis

*D. purpurea* is a medicinal plant that produces various cardiac and steroidal glycosides [6, 29]. The secondary metabolite biosynthesis pathways have not been sufficiently investigated so far; therefore, it would be very useful to find the secondary metabolite synthesis-involved hub genes in the context of metabolic engineering. Among 32341 unigenes derived from eight mentioned samples, 16185 filtered unigenes were merged with five major metabolites (S1 Table). Systems biology and integrative multi-omics studies provided an opportunity to study the important aspects of metabolic processes and complexities of the transcriptome and metabolome in non-model plants [14]. Combining the transcriptome with metabolome data could lead to an accurate network control of the biosynthesis of secondary metabolites. The WGCNA package was applied to find different modules of co-expressed genes and the correlation of the secondary metabolites with hub genes in each module [14].

Using WGCNA algorithms, the adjacency matrix was substituted with the weighted adjacency matrix by raising the correlations to the power of 16, which was selected through the scale-free topology criterion [35, 36]. The scale-free topology model was not improved after the power increase. At this power, the high mean number of connections was maintained;

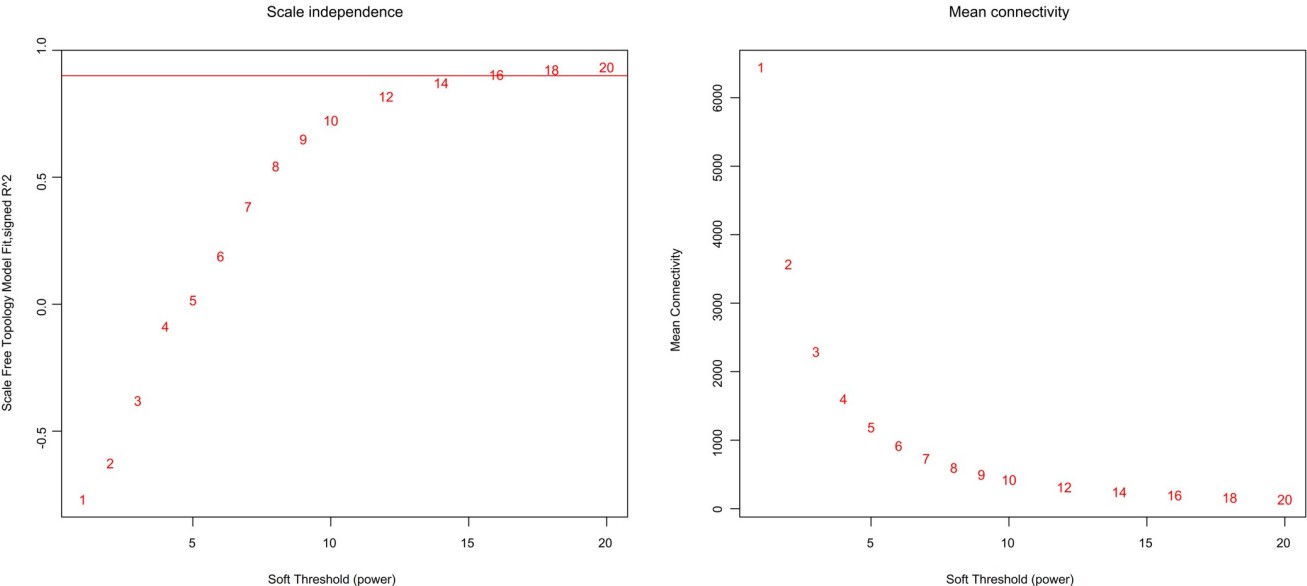

**Fig 2. Selection of an appropriate soft threshold power of β.** Left; the scale-free fit index of power (β) was estimated to be 16 based on the threshold limit of 0.9. Right; the mean connectivity versus soft-thresholding power.

moreover [36], the scale-free topology fit index reached to 0.9 at the mentioned power, which was selected to produce a hierarchical clustering tree (Fig 2).

The highly interconnected genes with similar expression profiles were clustered to a module based on TOM-based dissimilarity matrix [35, 36]. Therefore, the module network dendrogram was constructed through clustering Module Eigengene (ME) distances (Fig 3). First, 180 modules were identified, and then, 34 distinct modules were generated in different colors through dynamic tree cut and merged dynamic (Figs 3 and 4).

## Correlated modules with the secondary metabolites

Results showed that 34 modules were related to five major secondary metabolites based on the Pearson correlation coefficient and *P-value*. Higher correlation, significant *P-value*, with secondary metabolites profiles, were observed for seven modules including coral3, lightpink4, chocolate3, blue2, coral4, darkorange2, and lightsteelblue modules (Fig 5 and S2 Table).

The coral3 module showed a high positive correlation with strospeside and glucodigitoxin (Fig 5 and S2 Table); also, it was involved in the processes of terpenoid backbone biosynthesis,

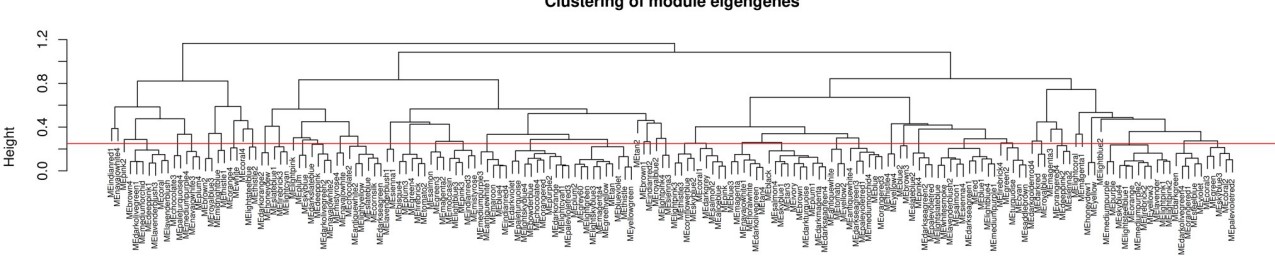

**Fig 3. Clustering of 180 modules.** The horizontal red line shows the height cut of 0.25, which corresponds to the correlation of 0.75, to merging the modules.

Cluster Dendrogram

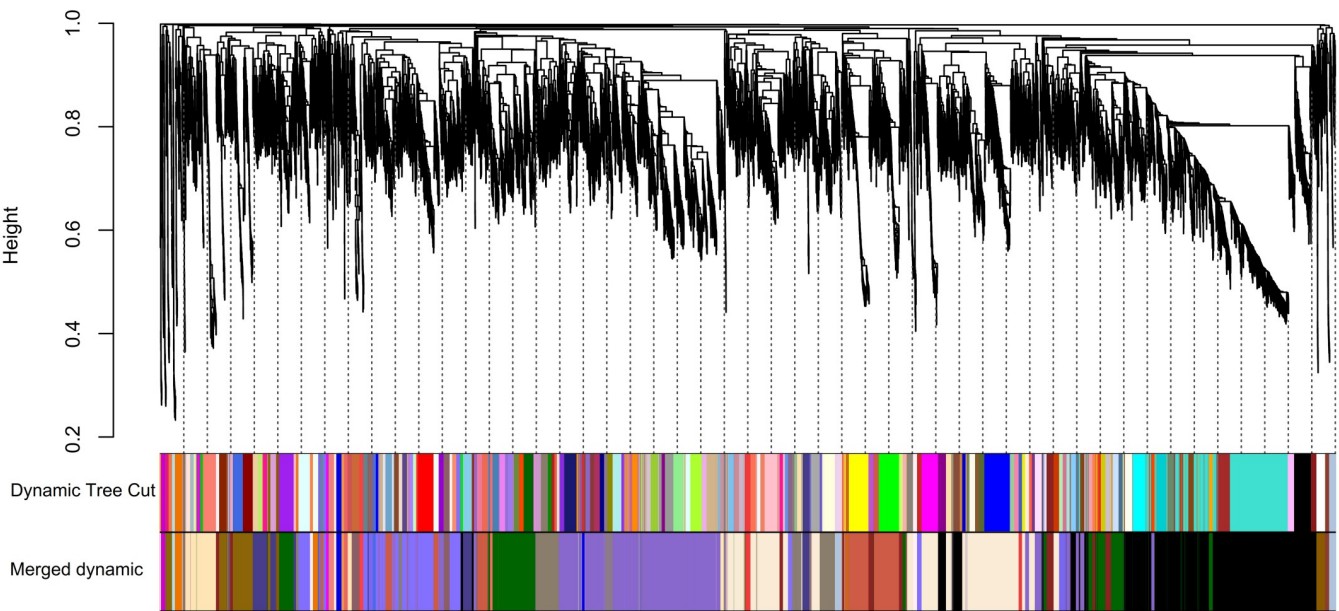

**Fig 4. Clustering of genes and modules.** The cluster dendrogram at the top of the plot shows co-expressed genes. The branches and color bands at the bottom of the plot represent the assigned module.

steroid biosynthesis, carotenoid biosynthesis, diterpenoid biosynthesis, flavonoid biosynthesis, flavone and flavonol biosynthesis, sesquiterpenoid, and triterpenoid biosynthesis (S3 Table). The darkorange2 and lightsteelblue modules were significantly associated with digitoxigenin bis-digitoxoside (Fig 5 and S2 Table) and involved steroid biosynthesis-related genes (S3 Table). The lightpink4 module significantly associated with glucodigitoxin (Fig 5 and S2 Table) was involved in steroid biosynthesis (S3 Table), and the chocolate3 module with high positive correlation and high significant level with digitoxigenin bis-digitoxoside (Fig 5 and S2 Table), was involved in terpenoid backbone biosynthesis (S2 Fig and S3 Table). It was found that the blue2 and coral4 modules were correlated with digitoxigenin bis-digitoxoside and gitoxin (Fig 5 and S2 Table). S3 Table represents data of some genes in selected modules involved in the process of secondary metabolites biosynthesis (S3 Table).

The MM and GS were calculated in order to investigate the correlations between individual genes and metabolites. As it could be observed in Fig 6, the strongest connectivity interaction is between chocolate3 module and digitoxigenin bis-digitoxoside (Fig 6). The chocolate3 module showed the most significant correlation between MM and GS (r = 0.5, p = 3.1e-06) (Fig 6). The high correlation between GS and MM indicated the considerable role of the modules key genes in the underlying biological functions of secondary metabolites synthesis. Functional annotation and enrichment analysis of chocolate3 indicated that three transcripts including *Plastid Terminal Oxidase* (*AT4G22260*, *IM*), *Isopentenyl Phosphate Kinase* (*AT1G26640*, *IPK*), and *Polypeptide 2* (*AT2G29090*, *ABAH2*) were involved in terpenoid metabolic (GO:0006721) and lipid biosynthetic processes (GO:0008610) (Fig 6).

The Module Eigengene adjacency was represented by hierarchical clustering and heatmap (Figs 7 and 8). A Module Eigengene summarizes the gene expression profile of each module. A dendrogram of the eigengenes and metabolites and a heatmap of their relationships were provided to evaluate the relationship between each module and digitoxigenin bis-digitoxoside

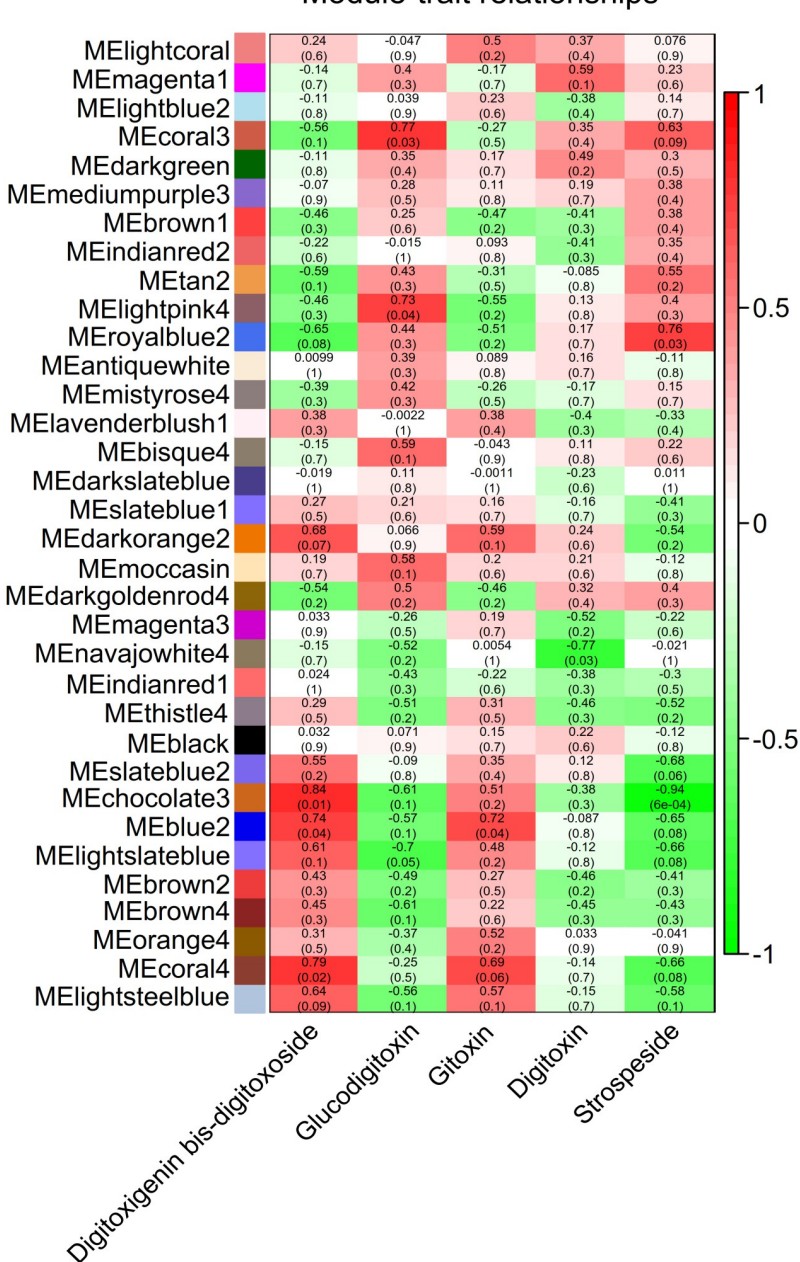

**Fig 5. Correlation of modules and secondary metabolites.** Module Eigengenes (MEs) and secondary metabolites are respectively represented by each row and column. Each cell contains the corresponding correlation at the top and *P-value* at the bottom. The positive correlation of the module with secondary metabolites and the negative correlation are respectively shown in red and green. The white spectrum indicates the inexistence of modules and secondary metabolites correlations.

(Figs 7 and 8). The dendrogram and heatmap indicated that chocolate3 module and digitoxigenin bis-digitoxoside are highly related (Figs 7 and 8). Conversely, the royalblue2 and tan2 modules are highly related, this meta-module is inversely correlated with digitoxigenin bis-digitoxoside (Figs 7 and 8). Squares of red color along the diagonal are the meta-modules.

## Module membership vs. gene significance cor=0.5, p=3.1e-06

**Fig 6. The scatterplot of Gene Significance (GS) for digitoxigenin bis-digitoxoside vs. Module Membership (MM) in chocolate3.** The high significant correlation between GS and MM for Digitoxigenin bis-digitoxoside in the chocolate3 module.

Eigengenes were represented by $I$, $J$ and etc.; for example, $E_J$ denotes the eigengene (module) of the $J^{th}$ module [39] (Fig 8).

**Gene ontology (GO) and KEGG pathway analysis.** To conduct the functional analysis of *D. purpurea* transcriptome, all unigenes of the candidate modules were selected to search against the TAIR database using BLASTX with *E-value* $\leq 10^{-5}$. Among the 1760 unigenes, only 1250 were protein-coding. Then, the functional annotation and enrichment analyses were carried out (Figs 9 and 10).

All of the above-mentioned modules' annotated unigenes were searched against the DAVID database to predict their functions. Using GO analysis, unigenes were classified into three distinct categories of BP, MF, and CC. Considering the BP category, GO term with the most genes was oxidation-reduction process (GO:0055114) (Fig 9). Jasmonic acid (JA) biosynthetic process (GO:0009695) is a key GO term shown in the GO analysis (Fig 9). JAs are recognized as signals in the plant stress responses, development processes, biosynthesis, and proper accumulation of secondary metabolites [45, 46]. In fact, they lead to a variety of biological responses in plants, such as defense responses to attacks by herbivorous insects or necrotrophic pathogens, biological responses to the injuries, increased production of secondary metabolites, male sterility, sex-determination of plants, and growth inhibition [47]. Moreover, JAs are associated with oxylipin biosynthetic process (GO:0031408), which constitute a family of oxylipins with the capability of inducing the expression of genes that code for enzymes catalyzing the formation of various secondary metabolites [48] (Fig 9). In addition, secondary

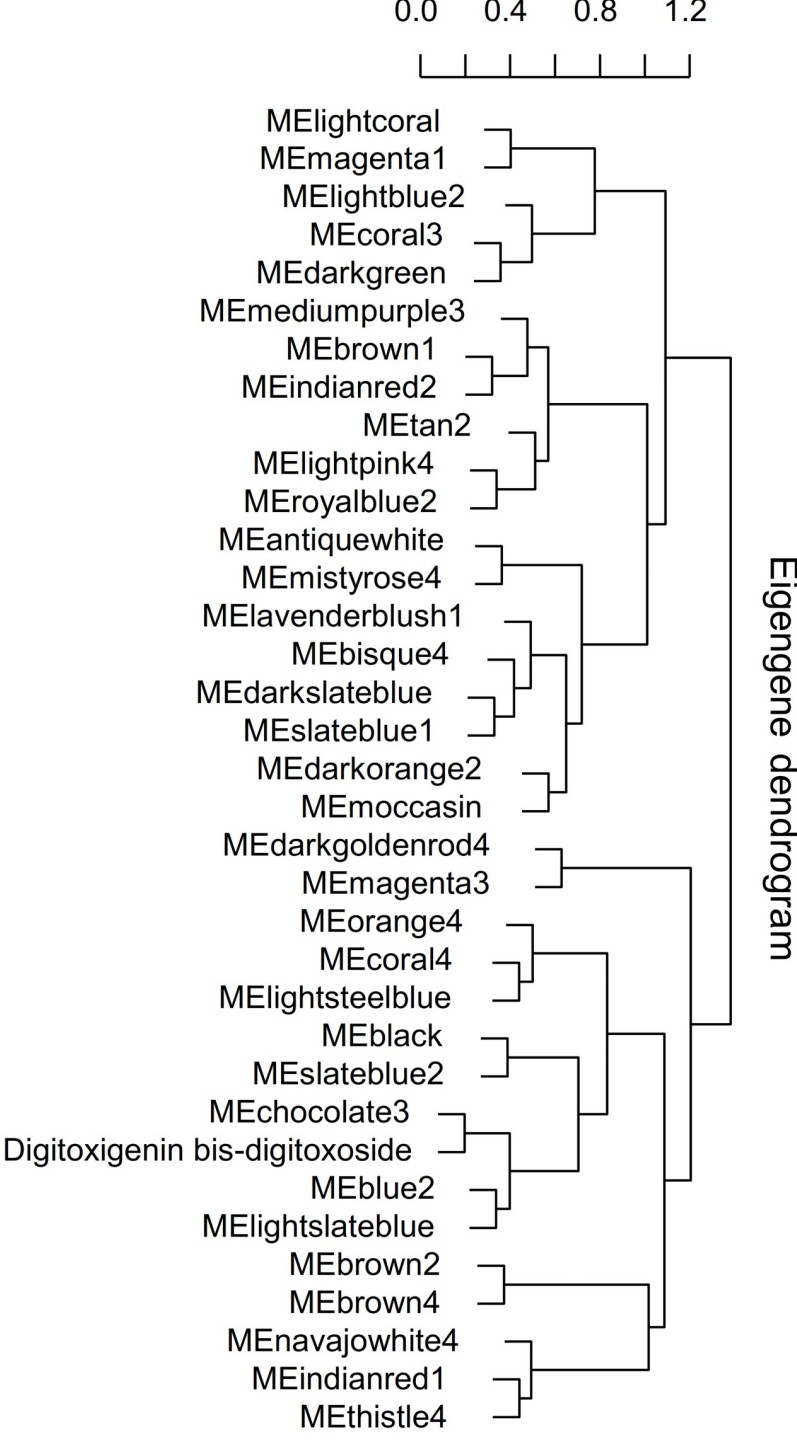

**Fig 7. Eigengene dendrogram of the modules and digitoxigenin bis-digitoxoside.** A hierarchical clustering dendrogram of eigengenes. The dissimilarity of $E_I$ and $E_J$ is shown by 1- cor ($E_I$; $E_J$).

metabolites act as defense molecules [46, 48]; therefore, the biosynthesis and proper accumulation of secondary metabolites lead to the plants defensive responses such as response to wounding (GO:0009611) (Fig 9). Also, they form defense proteins, such as proteinase inhibitors (PINs), in wounded tomato leaves with JA signaling pathways [45].

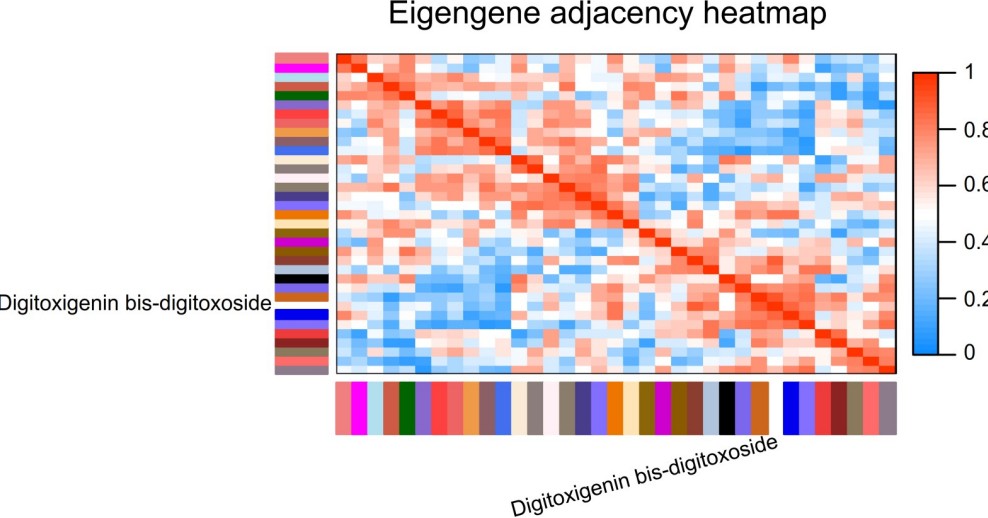

**Fig 8. The heatmap plot of adjacencies in the eigengene network, including digitoxigenin bis-digitoxoside.** Each row and column in the heatmap corresponds to one Module Eigengene (labeled by color) and or Digitoxigenin bis-digitoxoside. Low and high adjacencies (negative and positive correlations) are shown in blue and red. The connection strength (adjacency) between eigengenes $I$ and $J$ are defined as $A_{IJ} = (1 + cor(E_I; E_J))/2$.

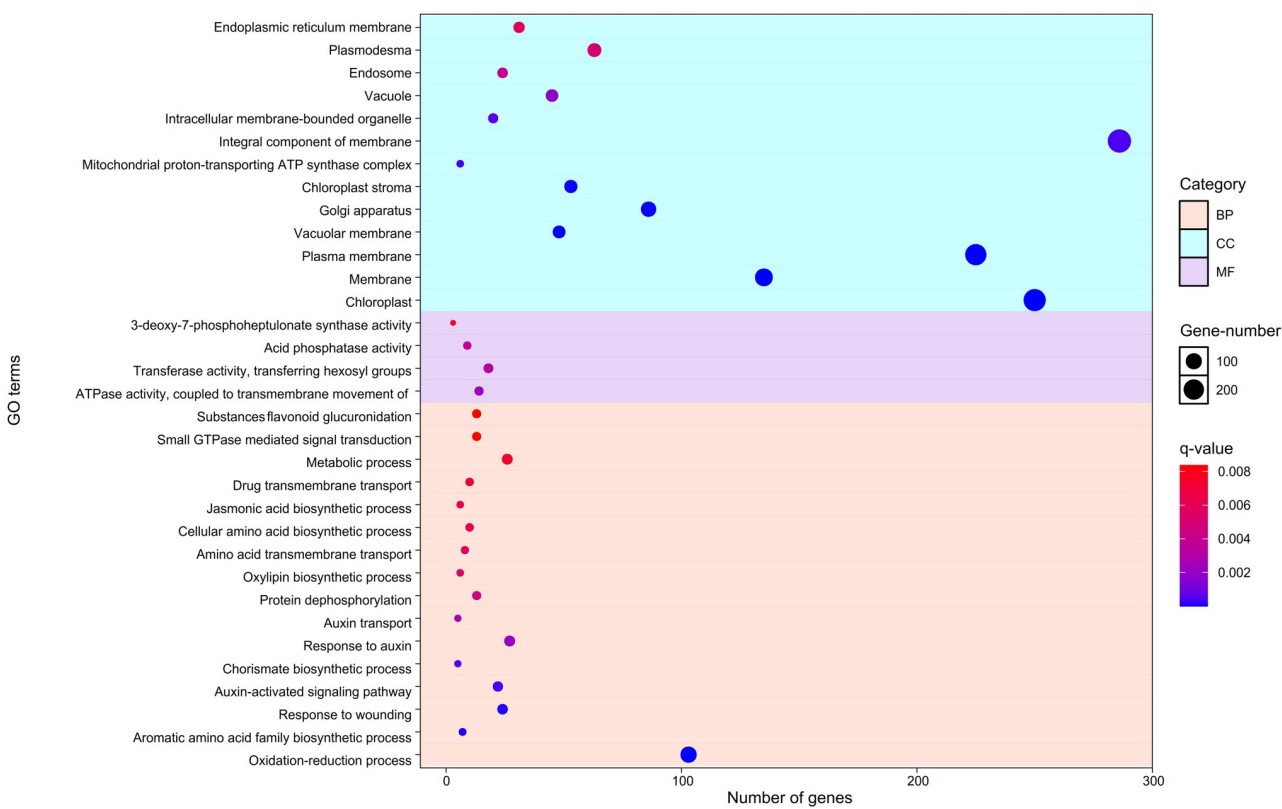

**Fig 9. The gene ontology analysis of candidate modules.** Gene ontology analysis of candidate modules classified into three functional categories including Biological Process, Molecular Function, and, Cellular Component.

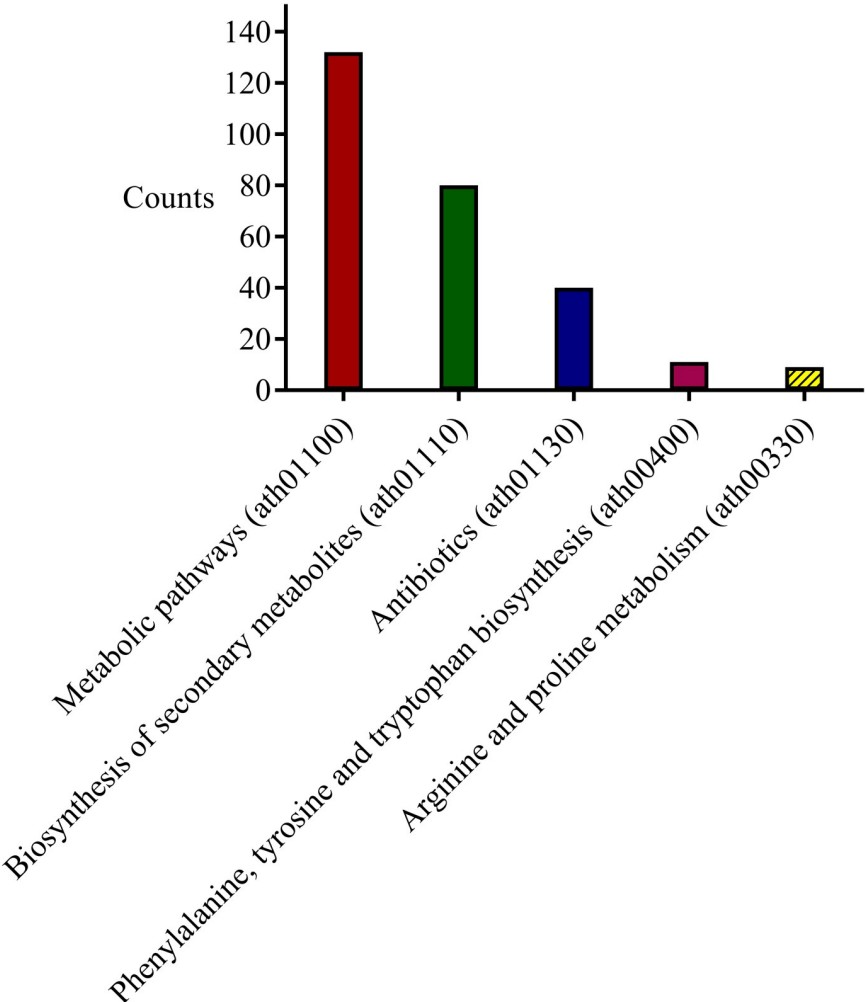

**Fig 10. The KEGG analysis of candidate modules.** KEGG analysis of candidate modules is performed by the DAVID database.

A number of secondary metabolites including nicotine, anthocyanins, glucosinolates, and terpenoid indole alkaloids (TIAs) are synthesized from proteinogenic amino acids [45]. Therefore, GO terms, such as aromatic amino acid family biosynthetic process (GO:0009073) and cellular amino acid biosynthetic process (GO:0008652) are associated with the biosynthesis of secondary metabolites [45] (Fig 9). Aromatic amino acid (AAA) family biosynthetic process (GO:0009073) is formed as a result of specific chemical reactions and pathways (Fig 9). It is noteworthy that AAAs include L-phenylalanine, L-tyrosine, and L-tryptophan involved in the protein synthesis [49, 50]. In addition, they are precursors for a wide range of secondary metabolites, various pigment compounds, and plant hormones, such as auxin and salicylate [49, 50]. In fact, AAA biosynthesis and degradation are considered as starting points for a large variety of secondary metabolites [50]. For example, tryptophan is a precursor for the synthesis of auxins, phytoalexins, glucosinolates, and alkaloids [49, 50]. Tyrosine is a direct precursor of coumarate in the phenylpropanoid pathway, as well as the synthesis of tyramine [49, 50] and meta-tyrosine, which is a non-proteogenic amino acid [50]. Tyrosine catabolism also leads to the synthesis of isoquinoline alkaloids, while phenylalanine is a precursor for a class of

sulfur-containing secondary metabolites, which is called phenylalanine glucosinolates, and volatile compounds including phenylpropanoids, benzenoids, phenylpropenes, and nitrogenous [49, 50]. The conversion of phenylalanine to cinnamate leads to its further metabolism to *p*-coumaroyl CoA [50]. It is involved in the stress-related mediating responses [50].

Auxin transport (GO:0060918), response to auxin (GO:0009733), and auxin-activated signaling pathway (GO:0009734) are considered as important GO terms (Fig 9). Tryptophan is a precursor to the family of auxin hormones [49, 50]. Indole-3-acetic acid (IAA) is the most abundant auxin required for almost all of the major developmental processes in plants including embryogenesis, seedling growth, root elongation, vascular patterning, gravitropism, and flower development [50]. Chorismate biosynthetic process (GO:0009423) is associated with AAA biosynthesis (Fig 9). In plants, chorismate is a precursor of AAA and a wide range of aromatic secondary metabolites [49, 50]. In addition, it is an initial compound for the biosynthesis of folates, such as tetrahydrofolate or vitamin B9, pigments, and isochorismate pathway to salicylate [49, 50].

Considering MF category, the most abundant of targets are enriched for ATPase activity to ATPase activity coupled to transmembrane movement of substances (GO:0042626) and transferase activity of transferring hexosyl groups (GO:0016758) (Fig 9). The former (GO:0042626) is associated with the ABC transporter mechanism; also, their mechanism is driven by ATP hydrolysis in order to act as the exporters and importers [51, 52] (Fig 9). It should be noted regarding the CC category that the most frequent targets are enriched for chloroplast (GO:0009507), plasma membrane (GO:0005886), and integral component of membrane (GO:0016021) (Fig 9). Most secondary metabolites stored in the vacuole are secreted to the apoplast [52]; moreover, secondary metabolites are produced in different subcellular compartments. Results showed that the biosynthetic pathway of secondary metabolites is consistent with the transport system; for example, terpenoids are transported by the G-type ABC transporter [52]. These transporters are localized in most plant cell membranes and classified as possibly vacuolar [51] (Fig 9). Therefore, the category of CC respectively includes 45, 48, 135, and 225 transcripts in vacuole (GO:0005773), vacuolar membrane (GO:0005774), membrane (GO:0016020), and plasma membrane (GO:0005886) (Fig 9).

The KEGG pathway enrichment analysis of candidate modules were conducted with the purpose of detecting the significant pathways. Results showed that there were five molecular pathways with *P-value* $\leq$ 0.01 detected using DAVID database [40] (Fig 10). Among these molecular pathways, the number of genes involved in metabolic pathways (ath01100), secondary metabolites biosynthesis (ath01110), antibiotics biosynthesis (ath01130), phenylalanine, tyrosine and tryptophan biosynthesis (ath00400), and arginine and proline metabolism (ath00330) were respectively determined to be 132, 80, 40, 11, and nine (Fig 10). The most important pathway was the secondary metabolites biosynthesis (ath01110) (Fig 10) in which genes were involved in terpenoid backbone biosynthesis (ath00900), sesquiterpenoid and triterpenoid (ath00909), steroid biosynthesis (ath00100), carotenoid biosynthesis (ath00906), diterpenoid biosynthesis (ath00904), flavonoid biosynthesis (ath00941), as well as flavone, and flavonol biosynthesis (ath00944) (S3 Table). The putative biosynthetic pathway of plant cardiac glycosides roughly comprised the biosynthesis of terpenoid backbone, steroid, and cardenolide [43] (S2 Fig). As an example, *Deoxy-D-xylulose-5-phosphate synthase* (*AT4G15560, DXS1*), *Geranyl diphosphate synthase 1* (*AT2G34630, GPS1*), *Delta14—sterol reductase* (*AT3G52940, FK*), and *DWF1* genes were involved in terpenoid backbone and steroid biosynthesis, respectively and in pairs (S2 Fig and S3 Table). All of the above-mentioned genes were identified in coral3 module that was associated with strospeside and glucodigitoxin (S2 Table). *HYD1* and *Sugar-Dependent 1* (*AT5G04040, SDP1*) were identified in the darkorange2 and lightsteelblue modules, respectively, which significantly were associated

with digitoxigenin bis-digitoxoside and were involved in the process of steroid biosynthesis (S2 Fig and S2 and S3 Tables).

**Identification of TFs.** TFs are key elements of plant metabolic engineering and regulatory proteins that improve the production of secondary metabolites [48, 53, 54]. Moreover, they regulate enzyme expression through integrating internal and external signals [53]. For example, a number of TF families including Apetal2/ethylene responsive factor (AP2/ERF), WRKY, basic helix-loop-helix (bHLH), basic leucine zipper (bZIP), MYB, and NAM, ATAF and CUC (NAC) are involved in biotic and abiotic stress responses through mediating biosynthesis and accumulation of secondary metabolites [55]. In the current study, TFs within candidate modules were identified and classified using iTAK database [41]. According to the observations, a total number of 89 TF-encoding genes belong to 30 families, including bHLH, MYB, bZIP, Cys2/His2-type (C2H2), AP2/ERF-ERF, C2C2-GATA, WRKY, GRAS, and others (S4 Table). In addition, C2H2 TF was identified in the six candidate modules (S4 Table). C2H2 TF (*mtfA*) regulates mycotoxin sterigmatocystin production and other secondary metabolism gene clusters, such as genes responsible for the synthesis of terrequinone and penicillin in *Aspergillus nidulans* [56, 57]. C2H2 Zinc-Finger family in drought, heat, and also salt responses in *Populus trichocarpa* [58].

Most of TF families, such as C2H2, AP2/ERF-ERF, and bHLH, contain gibberellic acid-mediated signaling pathway [14]; also, they are correlated with darkorange2, coral3, and light-pink4 modules (S4 Table). bHLH TFs can regulate the biosynthesis of the secondary metabolites including flavonoid, glucosinolates, isoquinoline alkaloid, nicotine alkaloid, diterpenoid phytoalexins, saponins, and anthocyanin [45, 48, 55]. The overexpression of bHLH TFs of *Triterpene Saponin Biosynthesis Activating Regulator1* (*TSAR1*) or *Triterpene Saponin Biosynthesis Activating Regulator2* (*TSAR2*) in *Medicago truncatula* increases the transcript levels of known triterpene saponin biosynthetic genes, as well as accumulation of triterpene saponins [55, 59]. It is noteworthy that bHLH and bZIP TF families regulate diterpenoid phytoalexins biosynthesis in *Oryza sativa*, which defend against the invasions of the blast pathogen [55]. The AP2/ERF-ERF and WRKY TF families act as regulatory proteins of *Catharanthus* terpenoid indole alkaloids and terpene biosynthesis [60]. In addition, JA-stimulated artemisinin biosynthesis within *Artemisia annua* is mediated by two AP2/ERF-type TFs, and *AaERF1* and *AaERF2*, as the overexpression of mentioned factors increases artemisinin accumulation in transgenic *A. annua* [61]. AP2/ERF TF family acts as a key regulator in the plant developments and stress responses [61]. The NAC TF family is involved in the anthocyanin accumulation within *Arabidopsis thaliana* and fruit crops [62, 63]. There was only one NAC TF observed in the darkorange2 module (S4 Table). NAC and bZIP TFs play vital roles in response to drought stress in *O. sativa* [64]. *PtrNAC72* in *Poncirus trifoliata* regulates putrescine biosynthesis [55], while *ANACO32* acts as a negative regulator of anthocyanin biosynthesis in *A. thaliana* [55]. *HbNAC1* is involved in latex biosynthesis and drought tolerance in *Hevea brasiliensis* [55]. Secondary metabolites, such as glucosinolates, flavonoids, Hydroxycinnamic acid amides (HCAAs), and proanthocynanins are also mediated by MYB proteins [55].

In the current study, GRAS, Teosinte branched 1, Cycloidea, Proliferating cell factors (TCP), and Trihelix families were identified as the hub genes. GRAS proteins play a key regulatory role in the plant development, abiotic stress, and phytochrome signaling [14] that are present in the blue2 and coral3 modules (S4 Table). GRAS TF family is the regulator of GA3 signaling and biosynthesis [14]; also, it interacts with *DWF1* (1.3.1.72) that is involved in the biosynthesis of steroids (ath00100). TCPs belong to the plant-specific bHLH TF family and are considered as key regulators of diverse developmental processes [65]. In *A. thaliana*, *mTCP3* expression induces the biosynthesis and accumulation of proanthocyanidins within endothelium and the outer seed coat layers; moreover, it activates many enzymatic and regulatory

genes involved in the flavonoid biosynthesis [65]. TCPs are also involved in the biosynthesis of plant hormones, such as brassinosteroids and jasmonic acids [65]. Trihelix transcription factors (TTFs), known as GT factors, are photoresponsive proteins that regulate environment-responsive secondary metabolisms [66]. *PatGT-1* acts as a negative regulator in the production of patchoulol through repressing genes in the pathway in *Pogostemon cablin* [66].

The coral3 module involves the maximum member of bZIP, GRAS, and MYB TF families (S4 Table). It is noteworthy that bZIP TF (*AabZIP1*) is involved in ABA signaling to regulate artemisinin biosynthesis in *A. annua* [67]. In addition, *ELONGATED HYPOCOTYL* (*HY5*), which is considered as a bZIP TF, plays a crucial role in the light-mediated transcriptional regulation of terpene synthase. For example, *AtTPS03* is involved in terpenoid biosynthesis in *A. thaliana* [68]. MYB, MYB-related, and forkhead-associated (FHA) TF families were involved in terpenoids and polyketides metabolisms in *Clinopodium chinense* [69]. MYB TFs are crucial for the terpenoid backbone biosynthesis [69].

**Identification of TRs.** As it was observed for TFs, the TRs can regulate the gene expression at the transcriptional level; moreover, their identification was carried out through iTAK database [41]. A total number of 26 TR-encoding genes from 11 families, such as mitochondrial transcription termination factors (mTERF), Auxin/indole-3-acetic acid (AUX/IAA), and GCN5-Related N-Acetyltransferases family (GNAT), was identified (S5 Table). Thirteen TRs were contained in coral3 module (S5 Table). In the current study, the most common TRs were AUX/IAA and mTERF (S5 Table). AUX/IAA family members were identified as short-lived nuclear proteins [70, 71], which could act as hub factors and regulate the gene expression in auxin signaling transduction [71]. According to an investigation carried out by Poutrain et al. (2011) on *Catharanthus roseus*, it was found that auxin negatively regulated the biosynthesis of monoterpenoid indole alkaloids (MIAs) and *CrIAA1* availability through a feedback mechanism [70]. In *Solanum lycopersicum* cv *MicroTom*, the positive regulation of mycorrhization and strigolactone biosynthesis was carried out by *Sl-IAA27* through regulating the expression level of *NODULATION SIGNALING PATHWAY1* (*NSP1*) [72]. In fact, AUX/IAA proteins are known as transcriptional repressors that mediate diverse physiological and developmental processes in plants; moreover, they are involved in the process of responding to stress [71, 73]. The mTERFs are key regulators of organellar gene expression (OGE) in mitochondria and chloroplasts, which can be implicated in all organellar gene expression steps ranging from the regulation of transcription to tRNAs maturation and regulation of translation [74]. In addition, mTERF proteins play a vital role in abiotic stress acclimation including excess light, UV-B exposure, heat, or altered salinity [74].

**Identification of PKs.** PKs add a phosphate group to certain amino acids of proteins. These phosphorylated proteins lead to the mechanisms of signal transduction that could improve the production of secondary metabolites [9]. In fact, PKs are identified as essential regulators of plant growth and development, including developmental patterning, hormone signaling, stress responses, and disease resistance [75–78]. In the seven modules, five families of PKs, including Receptor-like kinase-Pellel (RLK-Pelle), Cyclin-dependent kinases (CMGC), Calcium- and calmodulin-regulated kinase (CAMK), Sterility (STE), and Tyrosine kinase-like (TKL) PKs, were identified. It was also found that RLK-Pelle was involved in all selected modules (S6 Table). RLK-Pelle protein, which was considered as the largest family of PKs, played a key role in developmental processes of meristem proliferation regulation, organ specification, reproduction, and hormone signal transduction [79]. Also, it could act in the signaling networks that involved abiotic and biotic environmental stimuli [79]. Cowpea RLK-Pelle group allocated in the plasma membrane led us to achieve an understanding of the extracellular ligands and their role in activating the downstream pathways [80]. Moreover, it acted as a transmembrane protein with extracellular receptor domains and intracellular kinase domains

[80]. C2H2 TFs were associated with the promoter of the RLK-Pelle group. A number of C2H2 members were involved in the process of pathogen defense [80]. RLK-Pelle was recognized as the largest kinase group in the grapevine [81]. More than half of RLK-Pelle members were down-regulated in most of the tissues during the development, which indicated that they may have negative regulatory functions [81]. Several grapevine RLK-Pelle families were highly co-expressed, which suggested their possible interactions in the process of plant stress response signaling [81].

CAMK PKs were identified in the coral3 and lightsteelblue modules with four encoding genes (S6 Table). The grapevine CAMK (CAMK_CAMKL-CHK1) and TKL (TKL-Pl-4) proteins were up-regulated in response to stresses that caused dehydration, such as salt, PEG, and drought [81]. TKL PK was observed in coral3 module (S6 Table). The TKL family was found to be involved in the processes of growth, development, and stress response [82]. CMGC was also identified in the coral3 and darkorange2 modules with four encoding genes (S6 Table). According to the estimations, CMGC would localize the nucleus and cytoplasm. Moreover, it was down-regulated in response to salt, PEG, and drought treatments, while up-regulated in response to heat stress in the grapevine and cowpea [80, 81]. Many non-RLK groups, including CAMK, CMGC, STE, and TKL could positively regulate the plant growth [81].

**Identification of transporters.** Transporters are molecules essential for plant development that are involved in the plant transport system [83, 84]. In metabolic engineering, achieving a deep understanding of transport mechanisms and subcellular distribution of biosynthesized phytochemicals is crucial for the successful metabolic engineering of medicinal plants [85]. Using the TCDB database, 84 transporter families were identified (S7 Table). The ATP-binding cassette (ABC) transporter, Nitrate-peptide transporter (NRT), Multidrug and toxic compound extrusion (MATE), and Purine permease (PUP) are considered as transporters involved in the movement of secondary metabolites [52]. Secondary metabolites are transported in the intercellular, intracellular, and intratissue fashion [85]. An ABC family of C-type transporters is presumed in the vacuolar transport of anthocyanins [85]. NRTs transport nitrates and peptides, while MATE transporters act as proton antiporters [52]. In addition, a number of plant MATE transporters are involved in xenobiotic efflux, Fe translocation, Aluminium detoxification, and hormone signaling [52]. MATE transporters are involved in the vacuolar transporter for flavonoids and anthocyanin vacuolar transportation [85]. Most metabolites are transported through membranes, such as the carbohydrate transport in which the SWEET transporters lead to the nectar secretion, plant-microbe interaction, and embryo development [86]. Four transporters of the Sweet, PQ-loop, Saliva, and MtN3 family were identified in the candidate modules (S7 Table). In *Petunia axillaris*, *PaSWEET1* would supply sugar as energy for flowering and volatile biosynthesis [86]. The ABC proteins act as transporters of a diverse set of substrates including plant hormones, secondary metabolites, and lipid monomers [86]. We identified six transporters of the ABC superfamily (S7 Table). The mentioned proteins were localized in most plant cell membranes [51] and played a vital role in the process of plant development and defense mechanisms [84, 86]. G-type transporters ABC family was involved in the volatile release and formation of cutin, wax, and suberin [52, 86]. Terpenoids were transported by the G-type ABC transporters [52]. According to the findings, *Nicotiana plumbaginifolia pleiotropic drug resistance1* (*NpPDR1*) was a G-type ABC transporter involved in the resistance to fungal and oomycete pathogens through sclareol transmission to the plant surface [52]. *N. tabacum pleiotropic drug resistance1* (*NtPDR1*), which is a diterpene transporter, was also another G-type ABC transporter involved in the plant defense procedure through transporting various anti-fungal diterpenes, such as sclareol, manool, cembrene, and eucalyptol [52].

In our analysis, 27 transporters from the Mechanosensitive Calcium Channel (MCA) family were found (S7 Table). *MCA1* and *MCA2*, mechanosensitive calcium channels in *Arabidopsis*, were involved in a cold-induced increase in $[Ca^{2+}]_{cyt}$ and regulation of cold tolerance through a pathway other than the C-repeat binding factor/dehydration-responsive element binding-dependent (CBF/DREB1-dependent) pathway [87]. The Drug/Metabolite Transporter (DMT) superfamily with nine transporters, The Eukaryotic Nuclear Pore Complex (E-NPC) family with eight transporters, The Multidrug/Oligosaccharide-lipid/Polysaccharide (MOP) Flippase superfamily with seven transporters, The Domain of Unknown Function 3339 (DUF3339) family, The Major Facilitator Superfamily (MFS), The Amino Acid/Auxin Permease (AAAP) superfamily, The ATP-binding Cassette (YOP1) family and The $H^+$- or $Na^+$-translocating F-type, as well as V-type and A-type ATPase (F-ATPase) superfamily with six transporters, were the most identified and investigated transporters (S7 Table).

**Identification of mlncRNAs.** The mlncRNAs are considered as a subset of long noncoding RNAs (lncRNAs) and a new group of regulatory elements [88], which can be spliced, capped and polyadenylated [43]. In plants, mlncRNAs have regulatory roles in phosphate-starvation response, gender specific expression, nodulation, development, and response to hormones; also, they can affect cellular activity through specific sequences and RNA-folding structures [89]. A number of mlncRNAs play crucial roles in the organ development and defense responses through producing microRNAs; moreover, they have regulatory roles in plants through generating siRNAs [43]. It was found that the majority of *D. purpurea* mlncRNAs were species-specific and a considerable number of them showed tissue specific expression and involvement in plant development, cold and dehydration stress responses, and secondary metabolism [43].

Wu et al. (2012) confirmed that a number of mlncRNAs showed sense or antisense homology with protein-coding genes involved in secondary metabolism in *D. purpurea* [43]. For example, the 3' UTR of *D. purpurea 4-Hydroxy-3-methylbut-2-en-1-yl diphosphate synthase* (*HDS*) contained a 90 bp region with 87% identities to mlncR8. *HDS* was involved in the terpenoid backbone biosynthesis. In addition, the 5' UTR of *D. purpurea Solanesyl diphosphate synthase* (*SPS*) contained a 102 bp region that was highly complementary to mlncR31. *SPS* was involved in the biosynthesis of ubiquinone and plastoquinone [43]. In *Digitalis nervosa*, Salimi et al. (2018) showed that the expression levels of *Δ5–3β-hydroxysteroid dehydrogenase* (a key gene in cardenolides biosynthesis, *3β-HSD*), mlncRNA23, mlncRNA28, and mlncRNA30 across different tissues under normal conditions within leaves and roots were respectively high and low [88]. It indicated the adjacent relationship between *3β-HSD* and three mlncRNAs through an unknown mechanism. Also, it revealed that the possible accumulation of mentioned transcripts in the aerial parts of plants was associated with secondary metabolites biosynthesis site [88]. Due to the fact that the expression levels of three mlncRNAs and *3β-HSD* under stress conditions are similar and decreased, mlncRNAs are more possibly stress-responsive [88].

In the current study, four mlncRNAs were detected for coral3 module including JO461863 (mlncR13), JO460006 (mlncR6), JO466327, and JO462746 (S8 Table). Moreover, this module is associated with the secondary metabolites of glucodigitoxin and strospeside (S2 Table). The expression levels of mlncR6 were also higher in leaves and roots [43]. A region with significant similarity to the CDS of *SNF1-related protein kinase* (*SnRK*, JO461197) was included in mlncR6; also, expression of this gene was positively correlated with mlncR6 in cold and dehydration stresses [43].

## Identification of hub genes

Hub genes have regulatory roles and impact on the downstream pathways [12]. They are involved in basic processes, including the protein synthesis and secondary metabolite

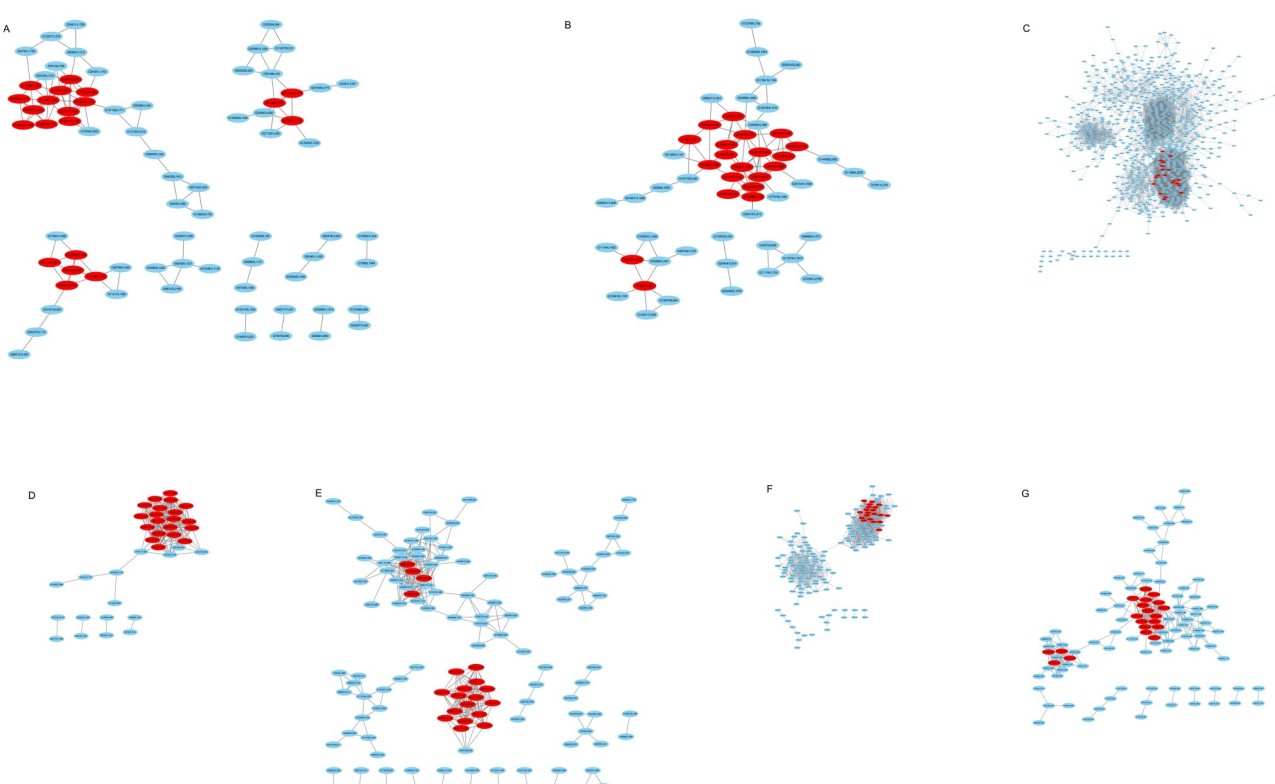

**Fig 11. A total number of 140 hub genes in seven modules.** The top 20 hub genes in each selected module, including blue2 (A), chocolate3 (B), coral3 (C), coral4 (D), darkorange2 (E), lightpink4 (F), and lightsteelblue (G) modules. Nodes represent genes in the network and red nodes indicate the hub genes. The gray line connecting two nodes indicates their connection.

biosynthesis. The hub gene *SCL14* was identified in the blue2 module, which was associated with the secondary metabolites of digitoxigenin bis-digitoxoside and gitoxin (S2 Table). This gene encodes a member of TFs GRAS family (Fig 11A and S9 Table). It is a TF of the scarecrow-like protein subfamily and interacts with TGA II TF that affects the transcription of stress-responsive genes. GRAS proteins are involved in gibberellic acid (GA) signaling, phytochrome A signal transduction detoxification, biotic, and abiotic stress-related response process, and development [90, 91]. This TF interacts with *DWF1*, which is involved in the biosynthesis of steroid (ath00100) and converts substances (S2 Fig). This enzyme was observed in coral3 module and showed a high positive correlation and significant *P-value* with glucodigitoxin and strospeside (S2 Table).

The hub gene *ATP-binding cassette G21* (*AT3G25620*, *ABCG21*) is related to the secondary metabolites of digitoxigenin bis-digitoxoside and gitoxin in the blue2 module (Fig 11A and S2 and S9 Tables). The hub gene *ATP-binding cassette G22* (*AT5G06530*, *ABCG22*) is related to the secondary metabolites of digitoxigenin bis-digitoxoside in the lightsteelblue module (Fig 11G and S2 and S9 Tables). The G-type ABC transporters are involved in the terpenoids transportation [52]. Another hub gene is *BIGPETAL* TF (*AT1G59640*, *BPE*), which controls the petal size and was identified in the blue2 module related to the secondary metabolites of digitoxigenin bis-digitoxoside and gitoxin (Fig 11A and S2 and S9 Tables). Furthermore, *TCP5* (*AT5G60970*) is a hub gene and encodes a TF within lightsteelblue module, which showed a correlation with digitoxigenin bis-digitoxoside (Fig 11G and S2 and S9 Tables). *MYB81*

(*AT2G26960*) is a hub gene and observed in chocolate3 module that was significantly associated with digitoxigenin bis-digitoxoside. The hub gene *Trihelix transcription factor GT-2* (*AT1G76890*, *GT-2*), which encoded a TF, was identified in the blue2 module that was associated with digitoxigenin bis-digitoxoside and gitoxin (Fig 11A and S2 and S9 Tables).

One of the key genes that produced the secondary metabolites of the above-mentioned medicinal plant within the lightsteelblue module was *UDP-glycosyltransferase 85A7* (*AT1G22340*, *UGT85A7*). This hub gene was related to the secondary metabolite biosynthesis of digitoxigenin bis-digitoxoside and cardenolides [43] (Fig 11G and S2 and S9 Tables). Among the hub genes that carry sugar compounds, *SWEET15* (*AT5G13170*) was identified in the lightsteelblue module, which was associated with digitoxigenin bis-digitoxoside (Fig 11G and S2 and S9 Tables). There is also another hub gene that acts as PKs, such as *ATAXIA-Telangiectasia Mutated* (*AT3G48190*, *ATM*). It was observed in the darkorange2 module significantly associated with digitoxigenin bis-digitoxoside (Fig 11E and S2 and S9 Tables). The hub gene *Cysteine-rich receptor-like protein kinase 8* (*AT4G23160*, *CRK8*) belongs to RLK-Pelle family, which was identified in the blue2 module that was associated with the secondary metabolites of digitoxigenin bis-digitoxoside and gitoxin (Fig 11A and S2 and S9 Tables). The hub gene *IPK* plays a vital role in terpenoid backbone biosynthesis (ath00900). It was identified in chocolate3 module which was significantly related to digitoxigenin bis-digitoxoside (Orthology: K06981) (Fig 11B and S1 Fig and S2 and S9 Tables). These hub genes were identified as promising candidates that improved the production of secondary metabolites. As it could be observed in the networks, several hub genes had the highest degree of connectivity among all 140 identified hub genes (Fig 11A–11G and S9 Table).

Fig 12 represents the interactions of hub proteins through STRING database and showed the minimum required interaction score on the highest confidence (0.900) (Fig 12 and S10 Table). According to the GO, their activities in response to stimulus (GO:0050896), cellular macromolecule metabolic process (GO:0044260), metabolic process (GO:0008152), and cellular process (GO:0009987) are respectively shown in red, blue, green, and yellow (Fig 12). The

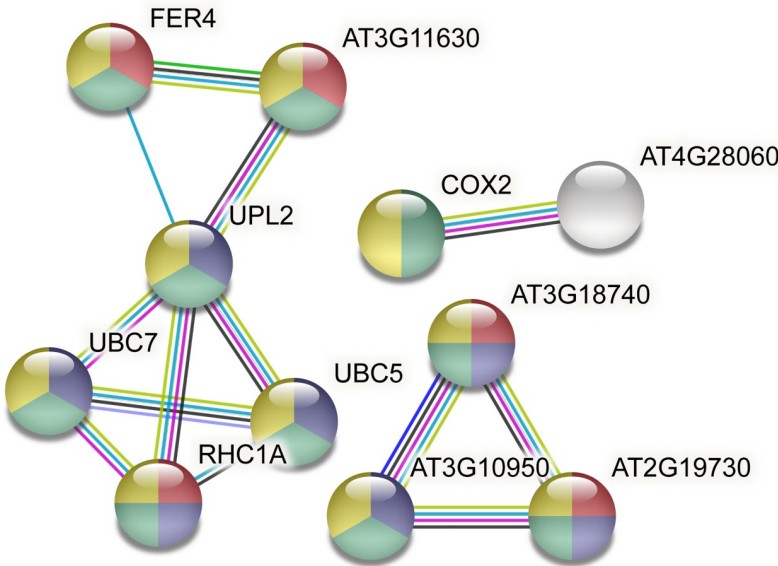

**Fig 12. The protein-protein interactions of hub proteins.** Response to the stimulus (GO:0050896), cellular macromolecule metabolic process (GO:0044260), metabolic process (GO:0008152), and cellular process (GO:0009987) are respectively shown in red, blue, green, and yellow.

interaction between nodes represented in Fig12 and related scores are provided in S10 Table (Fig 12 and S10 Table). Annotations of each node are also illustrated in (S11 Table).

## Validation of candidate genes

To investigate the induction effect of MeJA on *JAZ3*, *SCL14*, *DWF1*, and *HYD1* expression patterns, biennial plants were treated with 100 μM MeJA at the mentioned time points (Fig 13). In this study, *JAZ3* was chosen as a candidate gene to know whether it responds to MeJA induction or not. Jasmonates are the best recognized key signal transducers that stimulate the

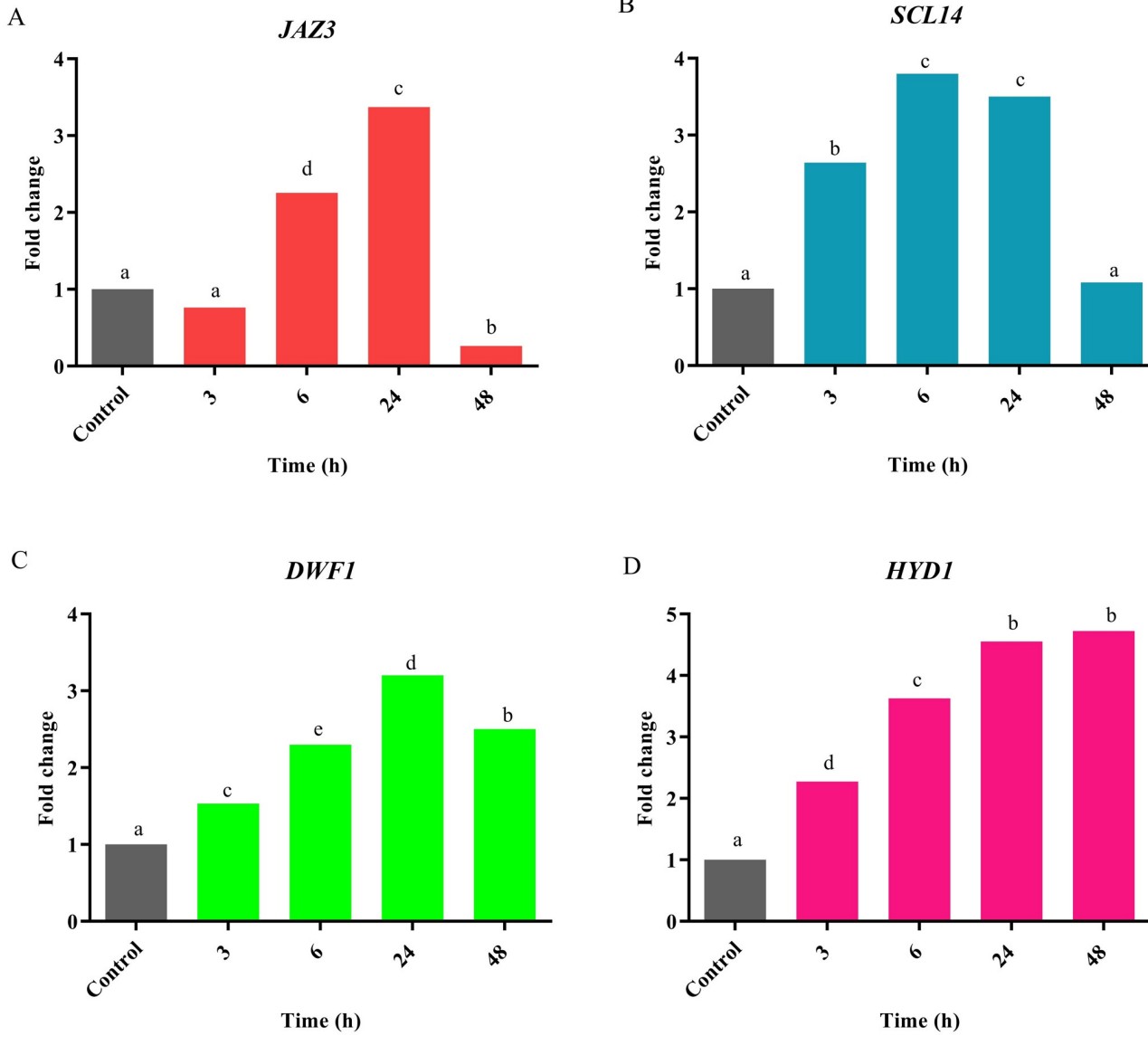

**Fig 13. The relative expression of candidate genes.** The expression pattern of *JAZ3*, *SCL14*, *DWF1*, and *HYD1* was evaluated under 100 μM MeJA treatment in *Digitalis purpurea* L. X axis represents the fold change of the expression value. Y axis represents time points. Vertical bars indicate ± SE of the mean (n = 3). The different letters on columns represent the significant difference given by Duncan's multiple range tests (*P-value* ≤ 0.05). There were no significant differences between equal letters (*P-value* ≤ 0.05). Although the expression of *JAZ3* is early induced, a significant suppression is observed after 48 h. Other key genes particularly *SCL14*, targeting *DWF1*, and *HYD1*, inducing cholesterol biosynthesis and cardiac glycoside content, showed a significant increase in the expression.

overproduction of secondary metabolites [92]. Pérez-Alonso et al. (2014) evaluated the elicitors including MeJA, induced production of cardenolide in *D. purpurea* L. [93]. They showed that digoxin and digitoxin contents were increased by 80 and 100 μM MeJA respectively [93]. Rad et al. (2022) evaluated the induction of secondary metabolites in *D. purpurea* by applying polyamines and MeJA in suspension cultures [94]. El-Sayed et al. (2019) confirmed that the highest Digoxin production was obtained by using malt extract autolysate medium supplemented by MeJA [95].

**Response of *JAZ3* to exogenous MeJA.** Although *JAZ3* showed no the significant expression at 3 h after MeJA treatment, it was significantly induced after 6 h and reached to the highest expression after 24 h, and decreased after 48 h (Fig 13A). This indicated that *JAZ3* was responsive to MeJA induction. The JAZ family are key transcriptional repressors in the JA signal transduction pathways [96–98]. Under normal conditions, JA content is maintained at a relatively low level, in which the JAZ repressor interacts with MYC2. This interaction inhibits downstream JA signaling responses including fertility process, root growth, senescence progress, secondary metabolite accumulation, and defense pathways. Large amounts of accumulated JA perceived by coronatine-insensitive 1(COI1) and JAZ proteins are degraded via the SCF$^{COI1}$ (Skp/Cullin/F-box) complex and leading to the activation of JA responsive genes and TFs (MYC2, MYC3, MYC4, MYB75, and others). In fact, COI1 binds the JA conjugate *JA-isoleucine* (*JA-Ile*) and this interaction enables the SCF$^{COI}$ complex to recognize JAZ proteins [96–98]. It is worth noting that, accumulation of JAs decreased after stress, and the expression of JA biosynthesis genes increased [99]. In the JA biosynthesis pathway, linolenic acid as the substrate for the biosynthesis of JA, convert to 12-oxo-phytodienoic acid (OPDA) by lipoxygenase (LOX), allene oxide synthase (AOS) and allene oxide cyclase (AOC). Then, OPDA is reduced by 12-oxo-phytodienoic acid reductase (OPR), to generate JA [45]. It seems that the increased expression of *JAZ3* at 6 h and 24 h after MeJA treatment is because of degradation of JAZ proteins and reduction of JA content (Fig 13A). Han and Luthe (2022) investigated the effect of response to Fall Armyworm (FAW) feeding on the accumulation of JA in caterpillar-resistant maize [99]. They showed the accumulation of JA continued until 1 h and then decreased at 6 h and 12 h. Moreover, JA biosynthesis genes in response to FAW feeding and exogenous application of MeJA were increased at 6 h in maize inbred lineTx601 and at 12 h in maize inbred line Mp708 [99].

Based on the following research, it is possible that the decrease in *JAZ3* expression 48 h after MeJA treatment will increase the secondary metabolites. Shi et al. (2016) found that overexpression of *SmJAZ3* in hairy roots produced lower levels of tanshinone whereas down-regulation of *SmJAZs* enhanced tanshione production [100]. Ju et al. (2019) discovered that the transcript abundance of *JAZ3* was rapidly up-regulated by MeJA treatment at 3 h, 6 h, and 12 h and was slowly and mildly up-regulated by ABA treatment at 24 h and finally the JAZ3 was rapidly and completely degraded by MeJA treatment in bread wheat and *Arabidopsis* [101]. Shoji et al. (2008) showed that jasmonate-induced up-regulation of nicotine biosynthesis is mediated by tobacco COI1 and JAZ repressors [102].

Li et al. (2021) confirmed that expression level of *SmJAZ3* was increased at 0.5 h and decreased at 1 h then increased from 2 h until reached to the highest level at 12 h after 100 μM MeJA elicitation in *S. miltiorrhiza* [103]. Han and Luthe (2022) showed the exogenous application of 0.01% MeJA was affected on the expression of *ZmJAZ3* as the highest level of expression was observed at 6 h after treatment in caterpillar-resistant maize [99]. In response to the exogenous application of MeJA, the highest expression of *ZmLOX*, *ZmAOS*, and *ZmOPR2* as JA biosynthesis, genes were observed at 6 h in maize inbred line Tx601 and the highest expression of *ZmLOX*, and *ZmOPR2* were detected at 12 h but the highest expression of *ZmAOS* was seen at 6 h in maize inbred line Mp708 [99].

According to GO analysis, JA biosynthetic (GO:0009695) and oxylipin biosynthetic (GO:0031408) processes are recognized as signals in response to the stresses and biosynthesis, and accumulation of secondary metabolites like TIAs, ginsenoside, taxol, MIAs, and artemisinin which are produce via biosynthesis of terpenoids and steroids pathway [45, 46, 48] that are according to our results as shown in Fig 9. In fact, it seems that the induced expression of *JAZ3* from 6 h to 24 h after MeJA elicitation is the cause of decrease in the accumulation of JAs and degradation of JAZ proteins at these point times (Fig 13A).

**Response of *SCL14* to exogenous MeJA.** Based on our results, the expression of *SCL14* encoding a GRAS TF was induced at 3 h and reached to the highest level at 6 h after 100 μM MeJA treatment and then decreased (Fig 13B). This gene is critical for indirect defense [104]. This gene is an additional component in the regulation and activation of some detoxification genes that are putatively involved in the detoxification of xenobiotics [105]. In fact, SCL14, TGA2, TGA5, and TGA6 are involved in the detoxification by inducing the expression of a subset of the genes. For example, the *scl14*, *tga2*, *tga5*, and *tga6* mutants in *Arabidopsis* were more susceptible to toxic doses of isonicotinic acid and 2,4,6-triiodobenzoic acid [105]. These genes control the majority of cis-jasmone-induced genes and cis-jasmone-induced defense [104]. In addition, MYC TFs as activators of the JA responsive signals, and TGA TFs as well-known players in the SA responsive signals and in detoxifying mechanisms but altogether, both of them mediate the responses to xenobiotics and their combined action is required for the complete activation of the responses [106]. Moreover, cis-acting element of activation sequence-1 (as-1) in the promoters of defense- and stress-related genes is mainly activated by the TGA II TF under auxin- and SA-mediated stimuli [105, 106]. SCL14 bind to the promoters containing (as-1)–like cis-elements too [106, 107]. Finally, under JA induction, MYC2 binds to the G-boxes and stringently requires the presence of TGA TFs at the as-1-like cis-acting element [108]. It seems that high expression of *SCL14* under MeJA treatment (Fig 13B) is because of degradation of JAZ3, and activation of MYC TFs (including MYC2, MYC3, and MYC4) and JA responsive genes. In addition, high expression of *SCL14* under MeJA treatment could be associated with the requirement for the presence of TGA TFs at the as-1-like element for MYC2 [108].

According to GO and KEGG pathway analysis, transport auxin (GO:0060918), auxin responsive genes (GO:0009733), and auxin-activated signaling pathways (GO:0009734) are considered as important GO terms (Fig 9) because of the activation of as-1 cis–acting element by the TGA II TFs responsive to auxin and SA mediated stimuli. Chorismate biosynthetic process (GO:0009423) (Fig 9) is associated with aromatic amino acid family biosynthetic process (GO:0009073) and phenylalanine, tyrosine, and tryptophan biosynthesis (ath00400) (Fig 10). In plants, chorismate is a precursor of AAA and an initial compound for isochorismate pathway to SA [49, 50]. In addition, AAAs are precursors for a wide range of secondary metabolites, various pigment compounds, and plant hormones, such as auxin and SA [49, 50]. Thus, high expression of *SCL14* can be related to the interaction with TGA TFs that are well-known players in the SA responsive signals (Fig 13B). *SCL14* interacts with *DWF1* which is involved in the biosynthesis of steroids (ath00100) including cholesterol that is a precursor of cardiac glycosides. In fact, in this pathway cholesterol is converted into the C21 steroid and cardiac glycoside via a series of enzyme catalytic reactions [109]. *SCL14* was associated with biosynthesis of the secondary metabolites of digitoxigenin bis-digitoxoside and gitoxin (S2 Table) and *DWF1* showed a high positive and significant correlation with glucodigitoxin and strospeside biosynthesis (S2 Table). It seems that the induced expression of *SCL14* under MeJA treatment (Fig 14) promotes the expression of *DWF1* (Fig 14), therefore enhances cholesterol biosynthesis as precursor of cardiac glycosides.

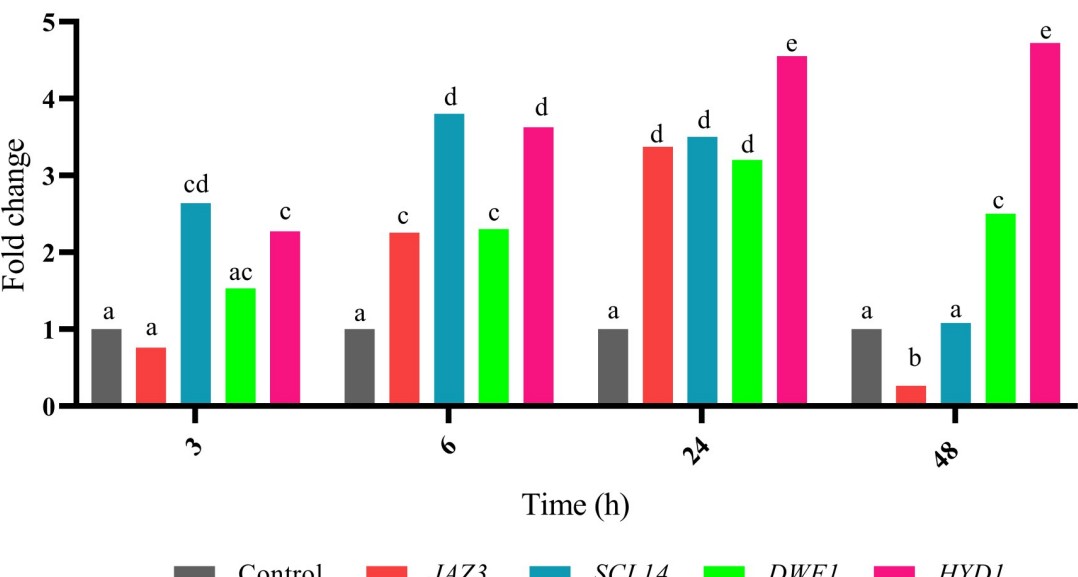

**Fig 14. The comparative relative expression of the candidate genes under 100 µM MeJA treatment in *Digitalis purpurea* L.** Here, the expression pattern of *JAZ3*, *SCL14*, *DWF1*, and *HYD1* is compared. X axis represents the fold change of the expression of candidate genes. Y axis represents time points. Vertical bars indicate ± SE of the mean (n = 3). The different letters on columns represent the significant difference given by Duncan's multiple range tests *(P-value* ≤ 0.05). There were no significant differences between equal letters *(P-value* ≤ 0.05). The interesting point is that the expression of *HYD1*, involved in the biosynthesis of cholesterol and subsequently the cardiac glycosides, shows a stable increase after 48 h. In contrast, other genes are suppressing and *JAZ3*, inducing the downstream genes, shows a significant suppression.

**Response of *DWF1* to exogenous MeJA.** The expression of *DWF1* increased after all time points as the highest expression was observed at 24 h (Fig 13C). This gene is downstream of SCL14 TF, therefore it targets *DWF1* (Fig 13C). It seems that high expression of *SCL14* and *DWF1* are associated with each other under MeJA treatment (Fig 14) and enhance cholesterol biosynthesis. Razdan et al. (2017) showed that the expression of *WsDWF1* increased at 6 h, 12 h, 24 h, 48 h, and 72 h after 0.1 mM MeJA treatment in *Withania somnifera* as the highest expression was observed at 48 h [110]. They confirmed that the highest withanolides accumulation was detected at 48 h [110]. Upadhyay et al. (2014) observed the *DWF1* accumulation at 5 h and decreased at 12 h under MeJA treatment in *Asparagus racemosus* [111]. Ciura et al. (2017) discovered that 100 µM MeJA treatment enhances the production of diosgenin in fenugreek (*Trigonella foenum-graecum*) tissues [112].

Wang et al. (2012) presented that overexpression of wild-type and mutant *BjHMGS1* (*3-hydroxy-3-methylglutaryl-CoA synthase 1*) in *Arabidopsis*, up-regulates the genes involved in sterol biosynthesis like *DWF1* [113]. *DWF1* is located downstream of *HMGS1*. *HMGS1* enhanced seed germination and sterol content, stress tolerance, reduced hydrogen peroxide ($H_2O_2$)–induced cell death, and increased resistance to *Botrytis cinerea* [113]. This study indicated that overexpression of upstream genes induces expression of downstream genes and enhanced secondary metabolite content. Mehmandoust Rad et al. (2020) showed that the application of polyamines and MeJA in *D. purpurea* had significant effects on the expression of *1-Deoxy-d-Xylulose 5-phosphate Reductoisomerase* (*DXR*) and contents of cardenolide and digitoxin and enhanced the expression of *DXR* and contents of cardenolide and digitoxin [114]. Thus, it is possible that by increasing the expression of *DWF1* under MeJA treatment, it increases the expression of the downstream genes like *3β-HSD* and increases the amount of cardiac glycosides. The expression of *3β-HSD* was increased under MeJA treatment and

pregnenolone was converted into progesterone [115]. In steroidal sapogenin biosynthesis pathway, *HYD1* converts 4α-methylcholesta-8,24-dien-3-ol to 4α-methylcholest-7,24-dien-3β-ol and *DWF1* converts desmosterol to cholesterol [112].

**Response of *HYD1* to exogenous MeJA.**   The expression of *HYD1* increased and showed the highest expression at 48 h after treatment (Fig 13D). Two genes, *DWF1* and *HYD1* are involved in the steroid biosynthesis pathway. It seems that high expression of *DWF1* and *HYD1* are associated with each other under MeJA treatment (Fig 14) so that enhances cholesterol biosynthesis. *HYD1* was associated with digitoxigenin bis-digitoxoside. Zhang et al. (2017) performed transcriptome analysis in leaves, roots, adventitious roots, and calli of *Periploca sepium* and identified higher expression of *HYD1*. Many genes like *HYD1* were significantly up-regulated in adventitious roots and calli but *DWF1* was significantly down-regulated in leaves [109]. Ciura et al. (2018) found that the highest diosgenin content was observed after treatment with 100 μM MeJA in *T. foenum-graecum* and the expression of the genes coding *HYD1* and *DWF1* was elevated [116]. Thus, it is possible that by increasing the expression of *HYD1* under MeJA treatment, it increases the expression of the downstream genes and increases the amount of cardiac glycosides.

According to GO and KEGG pathway analysis, endoplasmic reticulum membrane (GO:0005789) and integral component of membrane (GO:0016021) GO terms are associated with localization of *DWF1* [117]. The secondary metabolite biosynthesis (ath01110) (Fig 10) is related to the genes of *HYD1* and *DWF1* that are involved in the steroid biosynthesis pathway resulting in the formation of precursors of cardiac glycosides [109].

## Key genes and other genes effective on the cardiac glycosides production

In this study, WGCNA analysis has been used to identify key genes and pathways associated with biosynthesis of secondary metabolites in *D. purpurea* based on a systemic view and integration of metabolomics and transcriptomics data. Based on a systemic view, upstream genes were chosen as candidate genes. *DWF1* and *HYD1* were involved in the process of steroid biosynthesis to produce precursors of cardiac glycosides (cholesterol). *DWF1* is downstream of *SCL14*. In this study, the expression of *SCL14*, *DWF1*, and *HYD1* increased at all-time points. It seems that increasing the expression of these genes increases cholesterol as a precursor and the amount of cardiac glycosides is likely to increase. Cholesterol is the starting point of cardenolide formation in *Digitalis spp.* [31]. The expression of *3β-HSD* was increased at 1 h under 0.1 mM MeJA in *D. nervosa* [115]. *P5βR* and *P5βR2* were increased at 4 h and 8 h and decreased at 24 h and 48 h under 100 μM MeJA treatment in *D. purpurea* [118]. In cardenolide biosynthesis first, sterols transform to pregnenolone then, this pregnene is converted by *3β-HSD* into progesterone [118]. Progesterone converts to 5β-pregnan-3,20-dione by *Progesterone 5β-reductase* (*P5βR*) [118]. In this pathway, *P5βR2* catalyzes the 5β-reduction of the $\Delta^4$ double bond of several steroids [118]. Ordinal hydroxylations at C14 and C21 and lactone ring formation at C17 lead to aglycone digitoxigenin [118]. Finally, digitoxigenin bis-digitoxoside converts to digitoxin, which is a lipid soluble cardiac glycoside and precursor of glucodigitoxin and gitoxin [31, 119, 120]. Munkert et al. (2014) showed that MeJA treatment enhanced the transcription of *3β-HSDs* (*EcHSD2* and *EcHSD3*) and the accumulation of erysimoside and helveticoside in *Erysimum crepidifolium* [121].

Finally, an increase in the expression of upstream genes has been observed under MeJA treatment, and the amount of cardiac glycosides also increases [114]. Increasing the expression of *SCL14* as a hub gene, affects the whole network and induces the expression of *DWF1* and *HYD1*. Thus an increase in the expression of candidate genes from WGCNA analysis under methyl jasmonate treatment was expected and confirmed bioinformatic results.

## Conclusions

Cardiac glycosides are mainly generated through the members of *Digitalis* genus. In this study, for the first time, an attempt has been made to identify key genes and pathways associated with biosynthesis of secondary metabolites based on a systemic view and combination of transcriptome and metabolome data in *D. purpurea*. Based on the systems biology insight, the candidate genes could be effective in enhancing the production of secondary metabolites. MeJA treatment enhanced transcription of *JAZ3*, *HYD1*, *SCL14*, and *DWF1*. *DWF1* showed a high positive correlation with glucodigitoxin and strospeside and *HYD1* was associated with digitoxigenin bis-digitoxoside. Based on WGCNA, *SCL14* was a hub gene affecting on whole metabolite network under MeJA treatment. The key genes of *DWF1* and *HYD1* induce amount of precursors of cardiac glycosides. It is recommended that future research be carried out on the manipulation of metabolic pathways and metabolic engineering of the introduced genes to increase the production of valuable metabolites in *D. purpurea*.

## Supporting information

**S1 Fig. Two-year-old *Digitalis purpurea* pots.** The plants were treated with 100 μM MeJA (plus 0.1% Tween-20) in 0.1% ethanol. In addition, the controls were sprayed and watered with 0.1% Tween-20 in 0.1% ethanol. The leaf samples were collected at 3, 6, 24, and 48 hours after treatment.
(TIF)

**S2 Fig. Overview of cardiac glycosides biosynthesis pathway.** *IPK* with EC number 2.7.4.26 is shown in red box; is a hub gene and *DWF1* with EC number 1.3.1.72 is shown in red box, is a hub gene with a great role in compound conversion. Some genes shown in green box including *DXS1* with EC number 2.2.1.7, *GPS1* with EC number 2.5.1.1, *FK* with EC number 1.3.1.70, *HYD1* with EC number 5.3.3.5, and, *SDP1* with EC number 3.1.1.13 that were identified in selected modules showed key roles in biosynthesis of main secondary metabolites. All of these genes are involved in the biosynthetic pathway leading to the production of cardiac glycosides.
(TIF)

**S1 Table. The metabolome datasets of *D. purpurea*.** The metabolome datasets measured by Liquid Chromatography/Time-Of-Flight/Mass Spectrometry (LC/TOF/MS) method that were retrieved from Plant/Eukaryotic and Microbial Systems Resource database (MPR, http://metnetweb.gdcb.iastate.edu/PMR/).
(DOCX)

**S2 Table. The correlation and *P-value* of candidate modules associated with the secondary metabolites.**
(DOCX)

**S3 Table. Some genes in selected modules with central roles in biosynthesis of secondary metabolites.**
(DOCX)

**S4 Table. Transcription factors involved in the modules related to the production of secondary metabolites.**
(DOCX)

**S5 Table. Transcriptional regulators in the modules related to secondary metabolite production.**
(DOCX)

**S6 Table. Protein kinases involved to the biosynthesis of secondary metabolites.**
(DOCX)

**S7 Table. Protein transporters.** List of protein transporters in modules related to secondary metabolite production.
(DOCX)

**S8 Table. The key mlncRNAs identified in the coral3 module.**
(DOCX)

**S9 Table. The top 140 hub genes.** The top 20 hub genes in the networks of each selected module, ranked by the MCC method.
(DOCX)

**S10 Table. Protein-protein interactions of hub proteins.**
(DOCX)

**S11 Table. Annotation of hub protein interactions.** Annotation of each node in protein-protein interactions of hub proteins.
(DOCX)

## Acknowledgments

The authors would like to thank the Institute of Biotechnology for supporting this research and the Bioinformatics Research Group in the College of Agriculture (Shiraz University). We thank Doctor Ruhollah Naderi (Shiraz University) for kindly supplying seeds of barley. We thank Doctor Farzaneh Aram (Shiraz University) for her help in performing the RT-qPCR experiments.

## Author Contributions

**Conceptualization:** Ali Moghadam, Ahmad Tahmasebi.

**Data curation:** Ali Moghadam, Ahmad Tahmasebi, Ali Niazi.

**Formal analysis:** Fatemeh Amiri, Ali Moghadam, Ahmad Tahmasebi.

**Funding acquisition:** Ali Moghadam, Ahmad Tahmasebi.

**Investigation:** Ali Moghadam, Ahmad Tahmasebi, Ali Niazi.

**Methodology:** Fatemeh Amiri, Ali Moghadam, Ahmad Tahmasebi, Ali Niazi.

**Project administration:** Ali Moghadam, Ahmad Tahmasebi, Ali Niazi.

**Resources:** Ali Moghadam, Ahmad Tahmasebi, Ali Niazi.

**Software:** Fatemeh Amiri, Ali Moghadam, Ahmad Tahmasebi, Ali Niazi.

**Supervision:** Ali Moghadam, Ahmad Tahmasebi, Ali Niazi.

**Validation:** Ali Moghadam, Ahmad Tahmasebi, Ali Niazi.

**Visualization:** Ali Moghadam, Ahmad Tahmasebi, Ali Niazi.

**Writing – original draft:** Fatemeh Amiri.

**Writing – review & editing:** Ali Moghadam, Ahmad Tahmasebi, Ali Niazi.

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
