## [Decision Letter · Decision Letter 0]

21 Feb 2022

PONE-D-21-40451Identification of key genes involved in secondary metabolite biosynthesis in Digitalis purpureaPLOS ONE

Dear Dr. Moghadam,

Thank you for submitting your manuscript to PLOS ONE. After careful consideration, we feel that it has merit but does not fully meet PLOS ONE’s publication criteria as it currently stands. Therefore, we invite you to submit a revised version of the manuscript that addresses the points raised during the review process.

We look forward to receiving your revised manuscript.

Kind regards,

Mukesh Jain

Academic Editor

PLOS ONE

Journal Requirements:

3. We note that Figures 1 in your submission contain copyrighted images. All PLOS content is published under the Creative Commons Attribution License (CC BY 4.0), which means that the manuscript, images, and Supporting Information files will be freely available online, and any third party is permitted to access, download, copy, distribute, and use these materials in any way, even commercially, with proper attribution. For more information, see our copyright guidelines: http://journals.plos.org/plosone/s/licenses-and-copyright.

Reviewers' comments:

Reviewer's Responses to Questions

**Comments to the Author**

1. Is the manuscript technically sound, and do the data support the conclusions?

Reviewer #1: Yes

Reviewer #2: Yes

2. Has the statistical analysis been performed appropriately and rigorously? 

Reviewer #1: Yes

Reviewer #2: Yes

3. Have the authors made all data underlying the findings in their manuscript fully available?

Reviewer #1: No

Reviewer #2: Yes

4. Is the manuscript presented in an intelligible fashion and written in standard English?

Reviewer #1: Yes

Reviewer #2: Yes

5. Review Comments to the Author

Reviewer #1: The authors submitted an interesting and useful manuscript on understanding the biosynthetic pathway of bioactive metabolite formation in medicinally important plant Digitalis purpurea. This plant produced several medicinally important secondary metabolites. By comparing and correlating the metabolomics and transcriptomics data, the authors have identified, transcription factor families (30), transcriptional regulator families (11), transporters families (84), protein kinase families (5), four non-coding RNAs and, 20 hub genes involved in the biosynthesis of value-added secondary metabolites in D. purpurea. Authors find interesting result that they have identified some crucial genes such as SDP1, IPK, and TPS21 involve in the formation of steroid and terpenes. Further, the hub gene SCL14 which encodes GRAS transcription factor and Deata24-sterol reductase), are found to be involved in the biosynthesis of steroids. The study was performed methodically very well; I did not identify any significant problems with methodics or interpretation of the data. The study presents very clear data in a standard way and conclusions are well supported by the results.

Altogether, the study is credible, based on well-executed results. The manuscript is very well written and ready for publication. Provided supplementary data is adequate. I have only few important comments regarding the revision:

1. In page 2, line 17, 140 hub genes mentioned. Whereas in the very first page it was mentioned 20 hub genes, please check

2. Quality of some figures are really bad, please improve them, few figures are added in reverse orientation

3. I could not see Supplementary Table S1, the given RAR folder was corrupted.

Besides that, I feel that the manuscript is ready for publication and I support its acceptance.

Reviewer #2: 1- Why the authors have chosen so many tissue? Does secondary metabolites be expressed in all the tissues?

2- Authors must validate few candidate genes with real-time PCR analysis to confirm the RNA-seq data obtained form online resources.

3- Authors have just compiled the online based data. No study to confirm any dataset was done.

4- Tables does not need to put as main text. They can be supplied as supplementary data.

5- What is novel in the study?

6. PLOS authors have the option to publish the peer review history of their article (what does this mean?). If published, this will include your full peer review and any attached files.

Reviewer #1: No

Reviewer #2: No

---

## [Author Response · Author response to Decision Letter 0]

5 May 2022

Responses to comments of the Editor 

Answer

The manuscript was improved in accordance with the format of PLOS ONE. The main body, title page, and author affiliations are edited and corrected based on the format.

Answer

We thoroughly copyedited our manuscript for language usage, spelling, and grammar. In addition, an academic professional English editor edited our manuscript completely. Finally, we prepared a track-changed copy of our manuscript edited by a professional academic English editor as a separate file labeled 'New Manuscript' in supplementary information.

3. Upon resubmission, please provide the following: The name of the colleague or the details of the professional service that edited your manuscript

Answer

Our colleague Ms. Parisa Assayi (parisaassayi@email.com) who is an academic professional English editor paraphrased our manuscript and corrected it for language usage, spelling, and grammar.

4. A copy of your manuscript showing your changes by either highlighting them or using track changes (uploaded as a *supporting information* file). A clean copy of the edited manuscript (uploaded as the new *manuscript* file)

Answer

Finally, we prepared a track-changed copy of our manuscript edited by a professional academic English editor as a separate file labeled 'New Manuscript' in supplementary information.

5. We note that Figures 1 in your submission contain copyrighted images. All PLOS content is published under the Creative Commons Attribution License (CC BY 4.0), which means that the manuscript, images, and Supporting Information files will be freely available online, and any third party is permitted to access, download, copy, distribute, and use these materials in any way, even commercially, with proper attribution. For more information, see our copyright guidelines: http://journals.plos.org/plosone/s/licenses-and-copyright. We require you to either (1) present written permission from the copyright holder to publish these figures specifically under the CC BY 4.0 license, or (2) remove the figures from your submission:

Answer

We have removed the copyrighted section of Figure 1 from our submission and a new edited version was uploaded.

Answer

We have included captions for our Supporting Information files at the end of our manuscript in the part of Supporting Information.

Answer

We have rechecked our reference list and have complied with the Vancouver format. We have not cited any retracted reference. In addition, we did not add or remove any reference as all of editions to the reference list have been mentioned in the revised manuscript.

Responses to comments of the Reviewer #1

1. In page 2, line 17, 140 hub genes mentioned. Whereas in the very first page it was mentioned 20 hub genes, please check

Answer

Actually, we have chosen seven modules among 34 modules and identified top 20 hub genes in each of module. Therefore, a total of 140 hub genes were identified from these seven selected modules.

2. Quality of some figures are really bad, please improve them, few figures are added in reverse orientation

Answer

The quality of all figures was improved and corrected based on the format with PACE. The font inside the figures has also been set to Arial of 8-12. In addition, the orientation of the figures was changed based on the comment.

3. I could not see Supplementary Table S1, the given RAR folder was corrupted.

Answer

The S1 Table was presented in a new corrected Supplementary Information file. 

Responses to comments of the Reviewer #2

1- Why the authors have chosen so many tissue? Does secondary metabolites be expressed in all the tissues?

Answer

These tissues were required for performing WGCNA analysis. Therefore, more tissue types help us to find more reliable correlations for finding the genes effective on metabolite production. It makes it possible to obtain the common network genes involved in biosynthesis of secondary metabolites in different tissues. According to S1 Table, secondary metabolites be produced in all the tissues but in different amounts. 

2- Authors must validate few candidate genes with real-time PCR analysis to confirm the RNA-seq data obtained from online resources.

Answer

Thank you for your precise comment. Actually, we have tried a lot to complete this experiment but unfortunately we have been encountered with the COVID-19 pandemic and then our lab was forbidden for all of communications. Now days, we got into the financial troubles.

3- Authors have just compiled the online based data. No study to confirm any dataset was done.

Answer

Thank you for your comment. As mentioned in the previous comment we not succeed to confirm our in silico analysis in lab using real-time PCR analysis cause of the restrictions. On the other hand, we have tried to cover this defeat using other online famous and comprehensive databases. 

4- Tables does not need to put as main text. They can be supplied as supplementary data.

Answer

All tables have supplied as supplementary data.

5- What is novel in the study?

Answer

Cardiac glycosides are plant-made compounds mostly by members of the Digitalis genus, like Digitalis purpurea. D. purpurea (for digitoxin) is one of the major sources of the cardiac glycosides most frequently employed in medicine. In spite of the clinical significance of medicinal plants, hardly ever a comprehensive study on biosynthesis of metabolites has been considered. In this study, for the first time, an attempt has been made to identify key genes and pathways associated with biosynthesis of secondary metabolites based on a systemic view and combination of transcriptome and metabolome data.

---

## [Decision Letter · Decision Letter 1]

18 May 2022

PONE-D-21-40451R1Identification of key genes involved in secondary metabolite biosynthesis in Digitalis purpureaPLOS ONE

Dear Dr. Moghadam,

Thank you for submitting your manuscript to PLOS ONE. After careful consideration, we feel that it has merit but does not fully meet PLOS ONE’s publication criteria as it currently stands. Therefore, we invite you to submit a revised version of the manuscript that addresses the points raised during the review process.

We look forward to receiving your revised manuscript.

Kind regards,

Mukesh Jain

Academic Editor

PLOS ONE

Additional Editor Comments:

Authors need to address the comments of the Reviewer 2. I think the validation of a few genes via RT-qPCR does not require substantial funds. Thus, authors should validate at least a few of the important genes highlighted in the study.

Reviewers' comments:

Reviewer's Responses to Questions

**Comments to the Author**

1. If the authors have adequately addressed your comments raised in a previous round of review and you feel that this manuscript is now acceptable for publication, you may indicate that here to bypass the “Comments to the Author” section, enter your conflict of interest statement in the “Confidential to Editor” section, and submit your "Accept" recommendation.

Reviewer #1: All comments have been addressed

Reviewer #2: All comments have been addressed

2. Is the manuscript technically sound, and do the data support the conclusions?

Reviewer #1: Yes

Reviewer #2: No

3. Has the statistical analysis been performed appropriately and rigorously? 

Reviewer #1: Yes

Reviewer #2: No

4. Have the authors made all data underlying the findings in their manuscript fully available?

Reviewer #1: Yes

Reviewer #2: No

5. Is the manuscript presented in an intelligible fashion and written in standard English?

Reviewer #1: Yes

Reviewer #2: Yes

6. Review Comments to the Author

Reviewer #1: Authors have addressed all the queries raised in original revision. Now figures qualities have been improved. The manuscript can be accepted now.

Reviewer #2: The authors can take additional time to do the required essential experiments i.e., RT-PCR to confirm whether your plant behaves in the similar manner or act differentially.

7. PLOS authors have the option to publish the peer review history of their article (what does this mean?). If published, this will include your full peer review and any attached files.

Reviewer #1: **Yes: **DEBABRATA SIRCAR

Reviewer #2: No

---

## [Author Response · Author response to Decision Letter 1]

16 Sep 2022

Responses to additional comments of the Editor 

Authors need to address the comments of the Reviewer 2. I think the validation of a few genes via RT-qPCR does not require substantial funds. Thus, authors should validate at least a few of the important genes highlighted in the study.

Answer

Thank you for your comments. Actually, we validated some key genes using RT-qPCR and addressed the comments of the Reviewer 2.

Responses to the comments of the Reviewer #2

Reviewer #2: The authors can take additional time to do the required essential experiments i.e., RT-PCR to confirm whether your plant behaves in the similar manner or act differentially.

Answer

Thank you for your comment. Based on this comment, after integrative data analysis, we validated some candidate key genes that are related to the biosynthesis of main secondary metabolites using RT-qPCR.

---

## [Decision Letter · Decision Letter 2]

25 Oct 2022

Identification of key genes involved in secondary metabolite biosynthesis in Digitalis purpurea

PONE-D-21-40451R2

Dear Dr. Moghadam,

We’re pleased to inform you that your manuscript has been judged scientifically suitable for publication and will be formally accepted for publication once it meets all outstanding technical requirements.

Kind regards,

Mukesh Jain

Academic Editor

PLOS ONE

Additional Editor Comments (optional):

Reviewers' comments:

Reviewer's Responses to Questions

**Comments to the Author**

1. If the authors have adequately addressed your comments raised in a previous round of review and you feel that this manuscript is now acceptable for publication, you may indicate that here to bypass the “Comments to the Author” section, enter your conflict of interest statement in the “Confidential to Editor” section, and submit your "Accept" recommendation.

Reviewer #1: All comments have been addressed

Reviewer #2: All comments have been addressed

2. Is the manuscript technically sound, and do the data support the conclusions?

Reviewer #1: Yes

Reviewer #2: Yes

3. Has the statistical analysis been performed appropriately and rigorously? 

Reviewer #1: Yes

Reviewer #2: Yes

4. Have the authors made all data underlying the findings in their manuscript fully available?

Reviewer #1: Yes

Reviewer #2: Yes

5. Is the manuscript presented in an intelligible fashion and written in standard English?

Reviewer #1: Yes

Reviewer #2: Yes

6. Review Comments to the Author

Reviewer #1: Manuscript has been revised as per reviewers comments. Authors have addressed comments with qRT PCR validation of selected genes. With this revision, in my opinion this manuscript can be accepted.

Reviewer #2: In the article entitled "Identification of key genes involved in secondary metabolite biosynthesis in Digitalis

purpurea", The authors have done a good piece of work. I accept all the replies made by the authors to the reviewers comments

7. PLOS authors have the option to publish the peer review history of their article (what does this mean?). If published, this will include your full peer review and any attached files.

Reviewer #1: No

Reviewer #2: No

---

## [Editor Report · Acceptance letter]

2 Nov 2022

PONE-D-21-40451R2 

Identification of key genes involved in secondary metabolite biosynthesis in *Digitalis purpurea*

Dear Dr. Moghadam:

I'm pleased to inform you that your manuscript has been deemed suitable for publication in PLOS ONE. Congratulations! Your manuscript is now with our production department. 

Kind regards, 

on behalf of

Dr. Mukesh Jain 

Academic Editor

PLOS ONE